# Migratory and adhesive cues controlling innate-like lymphocyte surveillance of the pathogen-exposed surface of the lymph node

Yang Zhang[1], Theodore L Roth[1], Elizabeth E Gray[1†], Hsin Chen[1], Lauren B Rodda[1], Yin Liang[1], Patrick Ventura[2], Saul Villeda[2], Paul R Crocker[3,4], Jason G Cyster[1*]

[1]Department of Microbiology and Immunology, Howard Hughes Medical Institute, University of California, San Francisco, San Francisco, United States; [2]The Eli and Edythe Broad Center of Regeneration Medicine and Stem Cell Research, University of California, San Francisco, San Francisco, United States; [3]Division of Cell Signalling and Immunology, University of Dundee, Dundee, United Kingdom; [4]College of Life Sciences, University of Dundee, Dundee, United Kingdom

*For correspondence: jason.cyster@ucsf.edu

Present address: [†]Department of Immunology, University of Washington School of Medicine, Seattle, United States

Competing interests: The authors declare that no competing interests exist.

**Abstract** Lymph nodes (LNs) contain innate-like lymphocytes that survey the subcapsular sinus (SCS) and associated macrophages for pathogen entry. The factors promoting this surveillance behavior have not been defined. Here, we report that IL7R$^{hi}$Ccr6$^+$ lymphocytes in mouse LNs rapidly produce IL17 upon bacterial and fungal challenge. We show that these innate-like lymphocytes are mostly LN resident. Ccr6 is required for their accumulation near the SCS and for efficient IL17 induction. Migration into the SCS intrinsically requires S1pr1, whereas movement from the sinus into the parenchyma involves the integrin LFA1 and its ligand ICAM1. CD169, a sialic acid-binding lectin, helps retain the cells within the sinus, preventing their loss in lymph flow. These findings establish a role for Ccr6 in augmenting innate-like lymphocyte responses to lymph-borne pathogens, and they define requirements for cell movement between parenchyma and SCS in what we speculate is a program of immune surveillance that helps achieve LN barrier immunity.

## Introduction

Our ability to mount adaptive immune responses against skin-invading pathogens depends on the delivery of antigens to lymph nodes (LNs) for encounter by naive lymphocytes (*Cyster, 2010*; *Qi et al., 2014*). However, activation, clonal expansion and effector lymphocyte differentiation takes several days, whereas pathogens can undergo marked replication in a matter of hours. The skin contains immune effector cells that help keep pathogen replication in check, in a process referred to as 'barrier immunity' (*Belkaid and Segre, 2014*). Despite this, in many cases, intact pathogens travel within minutes via lymph fluid to draining LNs. Indeed some pathogens, such as *Yersinia pestis*, appear to have evolved to undergo marked expansion only after arrival in the LN (*St John et al., 2014*). Recently, there has been evidence indicating the existence of barrier immunity within LNs. The first LN cells exposed to lymph-borne antigens include the CD169$^+$ macrophages that extend between the subcapsular sinus (SCS) or medullary sinuses and the underlying parenchyma (*Barral et al., 2010*; *Iannacone et al., 2010*; *Phan et al., 2009*). Crosstalk between sinus-associated macrophages and IFNγ precommitted CD8 T cells and NK cells is important for mounting rapid Th1-like and NK cell responses against acute infection by *Pseudomonas aeroguinosa*, *Salmonella*

**eLife digest** The lymphatic system is a network of vessels and a vital part of our immune system. Amongst other things, the lymphatic system carries microbes that have entered the body – for example via to a cut or mosquito bite – to small, oval-shaped organs called lymph nodes. The lymph nodes are packed with immune cells that can be activated to help fight off infections, however certain microbes actually replicate inside the lymph nodes themselves.

Lymph nodes protect themselves from these infections by having some pre-armed immune cells that are ready to respond rapidly as soon as an invading microbe is detected. These cells, referred to as innate-like lymphocytes, position themselves at the exposed surfaces of the lymph node – the locations where microbes are most likely to enter the organ. However, it was not known which cues caused these immune cells to assemble and remain at these locations.

Zhang et al. now reveal that a signaling molecule called CCL20 attracts the innate-like lymphocytes to the lymph node's exposed surfaces, while a protein known as CD169 helps to securely attach the innate-like lymphocytes in place. Further experiments then confirmed that positioning the innate-like lymphocytes at this location made mice more able to fight off the disease-causing bacterium *Staphyloccus aureus*.

Unexpectedly, Zhang et al. also found that innate-like lymphocytes can move from the surfaces of lymph node through to the underlying tissue. This unusual migratory behavior might allow the lymphocytes to search a larger area for the infectious microbes, though further studies are needed to test this hypothesis. Future studies are also likely to focus on elucidating how the innate-like lymphocytes recognize different types of invaders, and how their activity keeps the lymph nodes healthy.

*typhimurium* and *Toxoplasma gondii* (*Coombes et al., 2012*; *Kastenmüller et al., 2012*). IL17 is a cytokine with roles in anti-bacterial and anti-fungal defense that is made abundantly by effector T cells at epithelial surfaces (*Littman and Rudensky, 2010*). Whether IL17 is produced rapidly during responses to subcapsular sinus-invaders in LNs is unclear.

In recent work, our group and others identified populations of innate-like (pre-formed effector) lymphocytes that are enriched near the SCS in peripheral LNs and are pre-committed to produce IL17 (*Do et al., 2010*; *Doisne et al., 2009*; *Gray et al., 2012*; *Roark et al., 2013*). These cells express high amounts of the chemokine receptors Ccr6 and Cxcr6 as well as the cytokine receptor IL7R, and they include a majority of $\alpha\beta$ T cells but also considerable numbers of $\gamma\delta$ T cells as well as non-T cells (*Gray et al., 2012*). Within the IL17-committed $\gamma\delta$ T cell population a major subset expresses a V$\gamma$4-containing TCR (according to the nomenclature of [*Heilig and Tonegawa, 1986*]), and undergoes expansion in response to challenge with imiquimod or complete Freund's adjuvant (*Gray et al., 2013*; *Ramirez-Valle et al., 2015*; *Roark et al., 2013*). In previous work, we found that innate-like lymphocytes isolated from peripheral LNs were heavily coated with CD169$^+$ macrophage-derived membrane fragments ('blebs') (*Gray et al., 2012*). This observation suggested there may be strong adhesive interactions between these cells and the CD169$^+$ macrophages. CD169 is the founding member of the Siglec family of sialic acid-binding lectins (*Crocker et al., 2007*; *Macauley et al., 2014*). Although CD169 is a defining feature of LN SCS macrophages and targeting antigens to CD169 can promote antibody responses (*Macauley et al., 2014*), the function of CD169 on these cells is not fully understood.

Pre-enrichment of innate-like lymphocytes near LN sinuses is thought to be important for allowing very rapid responses against lymph-borne invaders (*Gray et al., 2012*; *Kastenmüller et al., 2012*). Despite this, it is not known whether IL17-committed innate-like lymphocytes in LNs respond rapidly upon pathogen challenge, and little is understood about how these cells localize to or move in the subcapsular region. In this study, we found IL7R$\alpha^{hi}$Ccr6$^+$ innate-like lymphocytes were mostly LN resident and they produced IL17 within hours of bacterial or fungal challenge. Their proximity to the SCS was mediated by Ccr6 and was important for the rapid induction of IL17 following bacterial challenge. Real time intravital two photon microscopy and in vivo labeling procedures revealed that innate-like lymphocytes exchanged between the LN parenchyma and the SCS. Movement into the

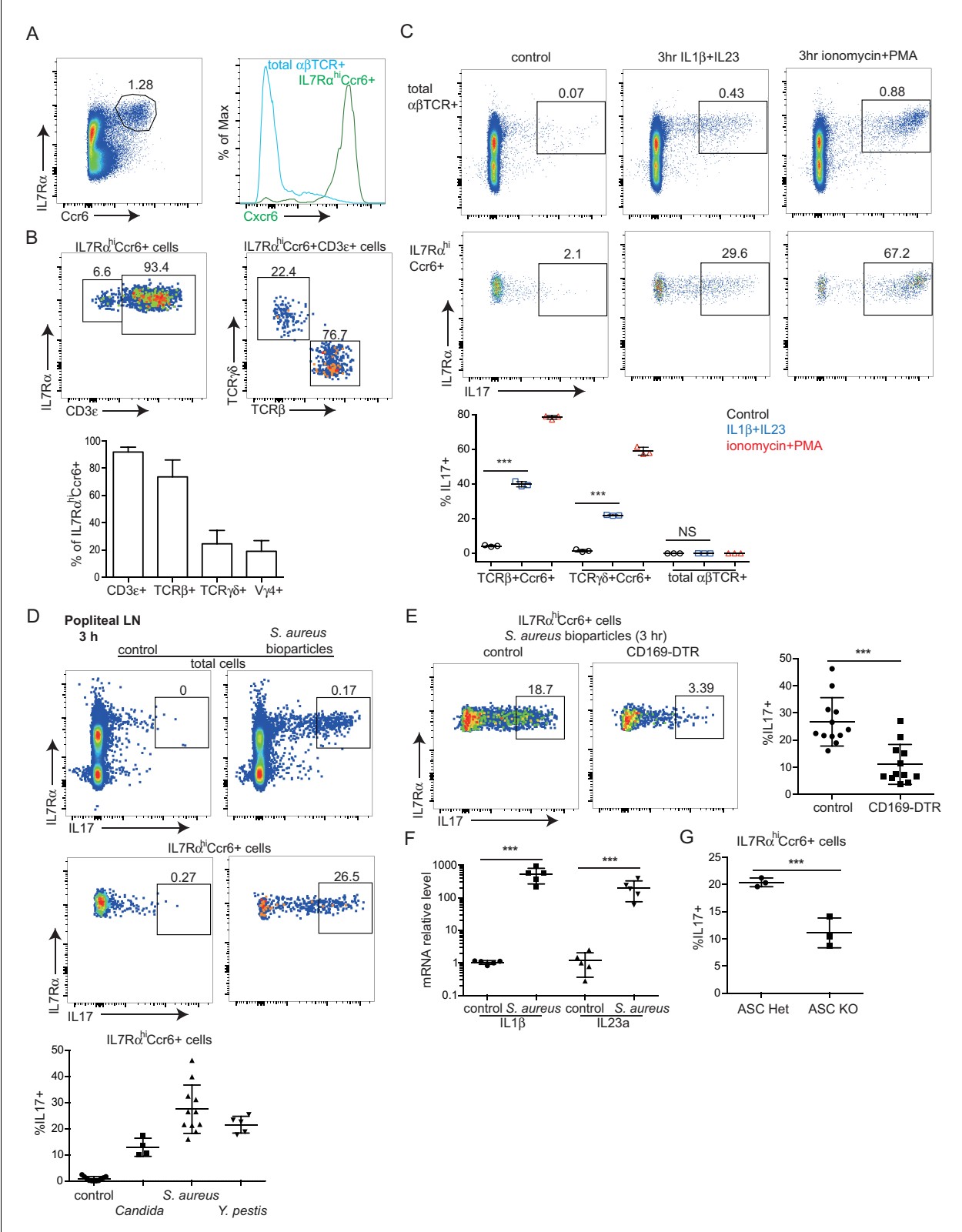

**Figure 1.** Rapid induction of IL17 expression by IL7Rα^hiCcr6^+ innate-like lymphocytes in a CD169+ macrophage-dependent manner following bacterial and fungal challenge. (**A**) Representative FACS plot showing IL7Rα^hiCcr6^+ staining of peripheral LN cells from a Cxcr6^GFP/+ mouse, and Cxcr6-GFP intensity on the gated cells. (**B**) Representative FACS plots showing CD3ε, TCRβ and TCRγδ staining of the IL7Rα^hiCcr6^+ population. Bar graph shows summary frequency data (mean ± sd) for more than 10 mice. (**C**) Intracellular FACS showing IL17 production among LN cells after 3 hr in vitro

*Figure 1 continued on next page*

Figure 1 continued

stimulation with IL1β and IL23 or phorbol 12-myristate 13-acetate and ionomycin. Graph shows summary data from 3 experiments. (**D**) Representative FACS plots showing IL17 production among popliteal LN cells 3 hr after footpad challenge with heat inactivated *C. albicans, S. aureus* coated bioparticles, and attenuated *Y. pestis*. Summary graph shows% IL17$^+$ cells among IL7Rα$^{hi}$Ccr6$^+$ cells. (**E**) IL17 production by IL7Rα$^{hi}$Ccr6$^+$ cells in control and CD169-DTR macrophage ablated mice treated with *S. aureus* bioparticles as in **D**, Summary graph shows% IL17$^+$ cells among IL7Rα$^{hi}$Ccr6$^+$ cells. (**F**) *Il1b* and *Il23a* mRNA level in popliteal LNs of *S. aureus* bioparticle challenged mice relative to controls, determined by qRT-PCR. (**G**) Summary graph to show% IL17$^+$ cells among IL7Rα$^{hi}$Ccr6$^+$ cells between control and ASC-deficient mice after 3 hr *S. aureus* bioparticle challenge. \*\*\*p<0.001 by student's t test. Data are representative of at least two experiments for panels **A**–**C**. Data are representative of two or more experiments with at least two mice per group for panels **D**–**G**. LN, Lymph node.

SCS was S1pr1 dependent, whereas return to the parenchyma involved LFA1 and ICAM1. Within the SCS, CD169-mediated adhesive interactions that helped retain the cells, presumably against the shear stresses exerted by lymph flow. This requirement was most prominent for the Vγ4$^+$γδ T cell subset of innate-like lymphocytes. These observations provide a model for understanding the mechanism by which innate-like lymphocytes survey the pathogen-exposed surface of the LN to protect the organ from infection.

## Results

### IL7Rα$^{hi}$Ccr6$^+$ innate-like lymphocytes near the SCS respond rapidly to pathogens

IL7Rα$^{hi}$Ccr6$^+$ innate-like lymphocytes within peripheral LNs express high amounts of Cxcr6 and they include ~70% αβ T cells, ~20% γδ T cells and 5–10% non-T cells (*Figure 1A,B*) (*Gray et al., 2012*). Consistent with previous findings, the IL7Rα$^{hi}$Ccr6$^+$gd T cell subset produced IL17 rapidly upon treatment with phorbol 12-myristate 13-acetate (PMA) and ionomycin or with the cytokines IL1β and IL23 (*Figure 1C*, lower graph) (*Cai et al., 2014*; *Gray et al., 2012*; *Gray et al., 2011*). These treatments also triggered rapid IL17 production from the IL7Rα$^{hi}$Ccr6$^+$αβ T cells (*Figure 1C*). We therefore tested whether both the αβ and γδ subsets of IL7Rα$^{hi}$Ccr6$^+$ T cells produce IL17 in skin draining LNs following bacterial or fungal challenge. IL17 is known to play a role in host defense against cutaneous *Candida albicans* and *Staphyloccus aureus* infection (*Cho et al., 2010*; *Conti and Gaffen, 2015*) and to be induced in rats by *Y. pestis* (*Comer et al., 2010*). Three hours after heat-killed *C. albicans, S. aureus* bioparticle, or attenuated *Y. pestis* footpad challenge, IL17 production in draining popliteal LNs was observed (*Figure 1D*). IL7Rα$^{hi}$Ccr6$^+$ lymphocytes were the dominant IL17 producers at this early time point after challenge (*Figure 1D*). The induction of IL17 expression in IL7Rα$^{hi}$Ccr6$^+$ cells upon bacterial challenge was dependent on CD169$^+$ SCS macrophages as the response was greatly blunted in CD169-DTR mice (*Miyake et al., 2007*) pretreated with DT to ablate these cells (*Figure 1E*). Since IL1β and IL23 induced IL17 production from IL7Rα$^{hi}$Ccr6$^+$ T cells in vitro, we looked for expression of these cytokines after *S. aureus* bioparticle challenge. Transcripts for both *Il1b* and *Il23a* were upregulated (*Figure 1F*). Induction of IL17 following *S. aureus* bioparticle challenge was compromised in mice lacking the ASC (Apoptosis-associated speck-like protein containing a CARD) inflammasome subunit (*Figure 1G*), consistent with a role for IL1β in activating IL7Rα$^{hi}$Ccr6$^+$ cells during the response to these bacteria.

### IL7Rα$^{hi}$Ccr6$^+$ innate-like lymphocytes are mostly LN resident

IL7Rα$^{hi}$Ccr6$^+$ lymphocytes make up ~0.5% of peripheral LN cells yet they represent only ~0.1% of cells in blood (*Figure 2A*), suggesting that the cells are largely non-recirculatory under homeostatic conditions. To test this more directly, we 'time stamped' cells in inguinal LNs of KikGR photoconvertible protein-expressing transgenic mice by brief violet light exposure (*Gray et al., 2013*). At 24 hr after photoconversion almost three quarters of the IL7Rα$^{hi}$Ccr6$^+$ cells and a similar fraction of the Vγ4$^+$Ccr6$^+$ cells remained resident in the LN, whereas more than 80% of the conventional αβ T cells and Ccr6$^-$ Vγ4$^+$ T cells had left the LN and been replaced by newly arriving cells. At 48 hr, the innate-like lymphocyte pool showed little further exchange, whereas naive αβ T cells were 90% replaced (*Figure 2B*). In a further approach, we examined the amount of cell exchange that occurred in parabiotic mice. Two weeks following surgery the naive αβ T cell compartment and the Ccr6$^-$

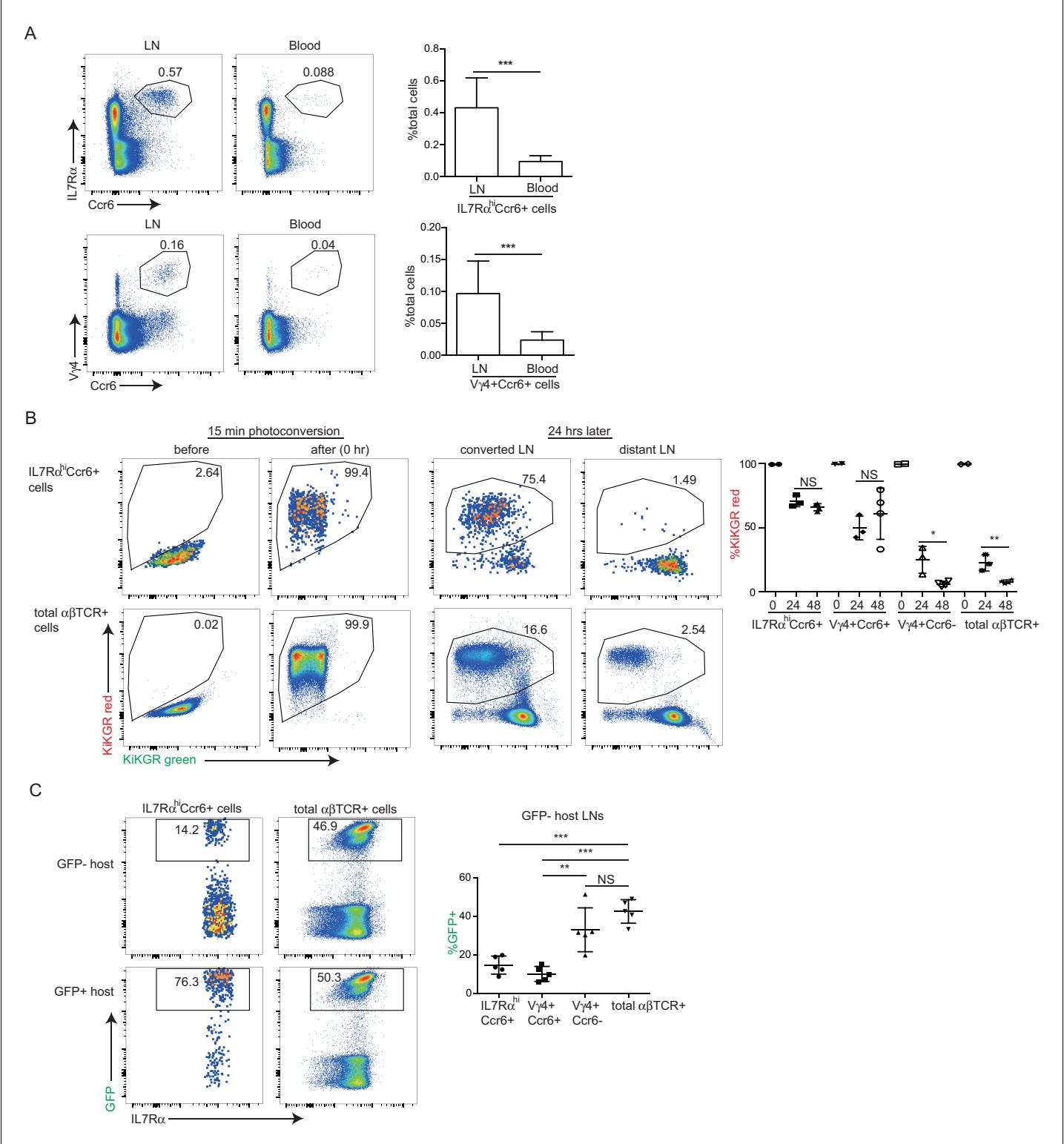

**Figure 2.** IL7Rα^hiCcr6^+ innate-like lymphocytes are mostly LN resident. (**A**) Representative FACS plots showing frequency of IL7Rα^hiCcr6^+ and Vγ4^+Ccr6^+ cells in LNs and blood. Graphs show summary data for more than 30 mice of each type. (**B**) FACS analysis of LN IL7Rα^hiCcr6^+ cells and naïve αβ T cells in KikGR mice before, immediately after and 24 and 48 hr after photoconversion. Summary data are pooled from three experiments and each point indicates an individual mouse. (**C**) FACS analysis of LN IL7Rα^hiCcr6^+ cells and naive αβ T cells in GFP-host and GFP+ host from parabiotic pairs. Summary data are pooled from two experiments and each point indicates an individual mouse. **p<0.01, ***p<0.001, by student's t test. Data are representative for at least two experiments. LN, Lymph node.

Vγ4$^+$ T cells had achieved full chimerism, whereas the innate-like lymphocytes showed only limited exchange between the paired mice (*Figure 2C*). Taken together, these findings indicate that most IL7Rα$^{hi}$Ccr6$^+$ cells are resident in the LN for multiple days and are not extensively recirculating under homeostatic conditions.

## Migration dynamics of innate-like lymphocytes at the SCS

Cxcr6-GFP is highly expressed by all IL7Rα$^{hi}$Ccr6$^+$ cells (*Figure 1A*), and in previous work, we observed that Cxcr6$^{GFP/+}$ cells migrate extensively in outer and inter-follicular regions, often in close association with CD169$^+$ SCS macrophages (*Gray et al., 2012*). A closer examination of Cxcr6$^{GFP/+}$ cell behavior in this region revealed that the cells frequently made contact with CD169$^+$ macrophages, and occasionally, lymphocytes could be observed crossing the layer of macrophages to reach the SCS (*Figure 3A* and *Videos 1* and *2*). Reciprocally, cells that were initially detected within the SCS could be observed migrating across the thick CD169$^+$ macrophage layer into the LN parenchyma (*Figure 3A* and *Videos 1* and *2*). To quantify the crossing events, cell tracks were generated automatically (*Figure 3—figure supplement 1*) and tracks crossing the SCS floor were manually enumerated. Among all the tracks of Cxcr6$^{GFP/+}$ cells within 50 μm of the capsule, ~25% were in the SCS (*Figure 3B*). Cell tracking analysis showed that about 3% of the cells in the SCS region traveled from the parenchyma into the sinus, and 3% of the cells traveled in the reverse direction, in the 30 min imaging periods (*Figure 3C*).

To quantitate the proportion of cells in the LN lymphatic sinuses at a given moment in time, we optimized an in vivo procedure to label lymph-exposed cells based on the established method of antibody pulse-labeling of blood-exposed cells (*Cinamon et al., 2008*). We targeted Thy1 since this marker is expressed by all the IL7Rα$^{hi}$Ccr6$^+$ cells while not being present on SCS macrophages (not shown). PE-conjugated antibody was used as the large size of PE (250 kD) reduces the rate at which the antibody accesses the lymphoid tissue parenchyma (*Pereira et al., 2009*). Thy1-PE antibody (0.2 μg) was injected into the footpad, the draining popliteal LN was isolated 5 min later and the frequency of labeled cells determined by flow cytometry. Approximately 10% of the total IL7Rα$^{hi}$Ccr6$^+$ cells were brightly labeled with Thy1-PE antibody compared with about 0.3% of naive T cells (*Figure 3D*). Among the innate-like lymphocytes, γδ T cells (predominantly Vγ4$^+$Ccr6$^+$ cells) were preferentially labeled, with around 15–20% of these cells being antibody exposed. By immunofluorescence microscopy, Thy1-PE-labeled CD3ε$^+$ cells were observed in the SCS and in nearby lymphatic sinuses, and few labeled cells were detected within the LN parenchyma, confirming that footpad injection of PE-conjugated antibody led to preferential labeling of lymph-exposed LN cells (*Figure 3E*). The broader labeling of the sinus by Thy-1 than by CD3 reflects the expression of Thy1 by lymphatic endothelial cells (*Jurisic et al., 2010*).

We also obtained information about lymphocyte-SCS macrophage proximity by following up on our finding that isolated innate-like lymphocytes are heavily coated with CD169$^+$ macrophage-derived membrane fragments ('blebs') (*Gray et al., 2012*). This coating is thought to occur at the time of LN cell dissociation, possibly because the SCS macrophages are tightly bound to the extracellular matrix and become fragmented during mechanical preparation of the tissue. Importantly, the blebs only become bound to cells that are associated with the macrophages at the time of isolation since co-preparation of LN cells from congenically distinct animals did not lead to cross acquisition of macrophage-derived blebs by innate-like lymphocytes from the different LNs (*Gray et al., 2012*). In accord with previous findings, 30–40% of the IL7Rα$^{hi}$Ccr6$^+$ innate-like lymphocytes isolated from control mice were CD169 macrophage-derived membrane bleb positive (*Figure 3F*). A higher frequency (40–55%) of the Vγ4$^+$γδ T cells were CD169 bleb positive, suggesting these cells may be preferentially associated with SCS macrophages (*Figure 3F*). Combining this analysis with Thy1-PE labeling showed that lymph-exposed innate-like lymphocytes, but not total αβ T cells, were enriched for CD169-bleb-positive cells (*Figure 3G*). The Thy1-PE$^+$ CD169$^-$ cells amongst total αβ T cells most likely correspond to recirculating cells that are exiting the LN via cortical and medullary sinuses.

## Ccr6 promotes innate-like lymphocyte positioning near the SCS

Given the high Ccr6 expression on the innate-like lymphocyte population, we asked whether this CCL20 receptor had a role in guiding innate-like lymphocytes to the subcapsular region. Although CCL20 is not abundantly expressed in LNs, it is expressed in LN lymphatic endothelial cells (LECs) at

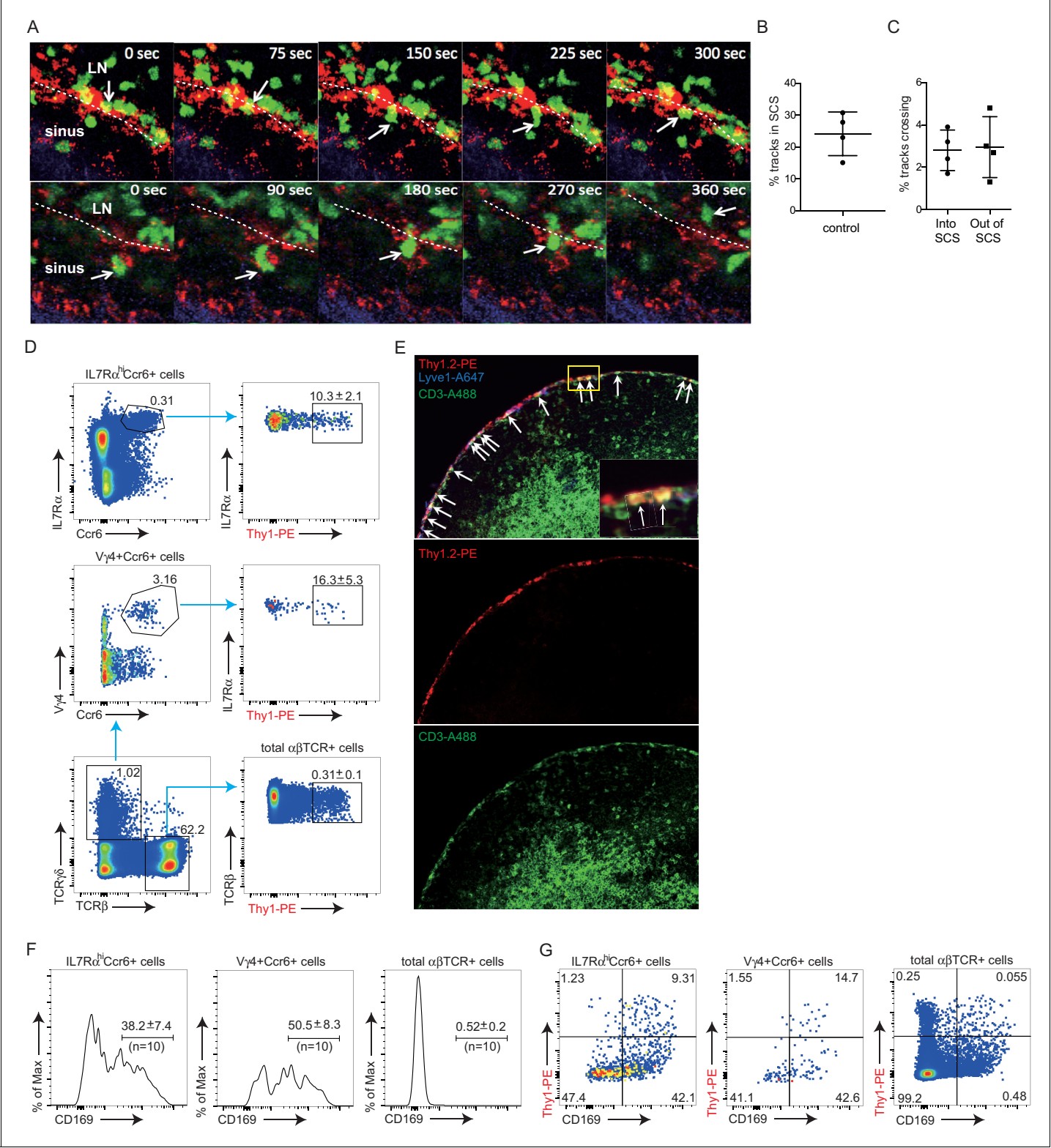

**Figure 3.** Migration dynamics, sinus exposure and CD169[+] macrophage interaction of LN innate-like lymphocytes. (**A**) Time series of Cxcr6[GFP/+] cell movement with respect to CD169[+] SCS macrophages. Upper panels: white arrow indicates a Cxcr6[GFP/+] lymphocyte in the LN parenchyma that crosses into the SCS. 300 s time series was taken from a 46 μm z stack. Lower panel: white arrow indicates a Cxcr6[GFP/+] lymphocyte that begins in the SCS and crosses into the LN parenchyma. 360 s time series was taken from a 34 μm z stack. Green, Cxcr6-GFP[+] lymphocytes; Red, CD169[+] macrophages; Blue, second harmonic. White dashed line indicates boundary between SCS and LN parenchyma. (**B, C**) Percent tracks in SCS compartment among the total

*Figure 3 continued on next page*

*Figure 3 continued*

tracks enumerated (**B**) and frequency of tracks crossing from the parenchyma into the SCS or out of the SCS into the parenchyma (**C**) in Cxcr6$^{GFP/+}$ control mice. Each point represents data from a single movie (two independent experiments). (**D**) In vivo 5 min Thy1-PE labeling of IL7Rα$^{hi}$Ccr6$^+$ cells, Vγ4$^+$Ccr6$^+$ cells and αβ T cells, analyzed by flow cytometry. Data are representative of at least 10 mice. (**E**) In vivo Thy1-PE labeling of cells analyzed in tissue sections. Costaining was with CD3-A488 (green) and Lyve1-A647 (blue). White arrows point out Thy1 and CD3 costained cells. (**F**) Frequency of IL7Rα$^{hi}$Ccr6$^+$, Vγ4$^+$Ccr6$^+$ and naive αβ T cells positive for CD169. Data are representative of 10 mice. (**G**) In vivo Thy1-PE labeling and CD169 staining on IL7Rα$^{hi}$Ccr6$^+$, Vγ4$^+$Ccr6$^+$ and naive αβ T cells, analyzed by flow cytometry. Data are representative of at least two experiments in each panel.

The following figure supplement is available for figure 3:

**Figure supplement 1.** Example of automatically generated tracks for Cxcr6$^+$ cells in a Cxcr6$^{GFP/+}$ mouse LN.

levels more than 100-fold higher than other LN lymphoid stromal cells (***Figure 4A***). Immunofluorescence microscopy showed evidence of CCL20 protein in the subcapsular sinus region (overlying follicular and interfollicular regions) but not in the medullary sinus region (***Figure 4B***, ***Figure 4—figure supplement 1***) consistent with findings in primate LNs (***Choi et al., 2003***; ***Pegu et al., 2007***). Innate-like lymphocytes were responsive to CCL20 by in vitro migration assays (***Figure 4C***). When CCL20 was injected subcutaneously into Cxcr6$^{GFP/+}$ mice, IL7Rα$^{hi}$Ccr6$^+$Cxcr6$^{hi}$ lymphocytes became clustered near and within the SCS in the draining LNs (***Figure 4D***, ***Figure 4—figure supplement 2***). Cells from these LNs showed reduced surface Ccr6, and increased CD169 staining and Thy1 labeling, consistent with their having been exposed to increased amounts of CCL20 and localizing near and within the SCS (***Figure 4E***).

We next examined the distribution of Ccr6-deficient (KO) cells in Ccr6$^{GFP/GFP}$ mice and found that the cells were reduced in density near the SCS (***Figure 4F***, ***Figure 4—figure supplement 3***) despite being slightly increased in total frequency in the LN (not shown). Instead, the cells were often distributed along the B-T zone interface (***Figure 4F***, ***Figure 4—figure supplement 3***). In Thy1-PE labeling experiments, Ccr6 KO mice showed reduced frequencies of labeled cells, a result that was most significant for the Vγ4$^+$ population (***Figure 4G***). Consistent with reduced proximity to SCS macrophages, Ccr6-GFP$^+$ IL7Rα$^{hi}$ cells and the Vγ4$^+$ subset from Ccr6 KO mice were associated with less CD169$^+$ blebs compared with Ccr6-sufficient controls (***Figure 4H***). The CD169$^+$ macrophage population in LN sections was unaffected by Ccr6-deficiency (not shown).

Given that the Vγ4$^+$Ccr6$^+$ cell population was most dependent on Ccr6 for Thy1 labeling and macrophage bleb acquisition (***Figure 4G,H***), we further studied the properties of these cells. Vγ4$^+$-Ccr6$^+$ T cells uniquely express the surface marker Scart2 (***Figure 4I***) (***Gray et al., 2013***; ***Kisielow et al., 2008***). In a complementary approach to Thy1 labeling, Scart2 antibody treatment was found to label fewer Vγ4$^+$ cells in LNs from Ccr6-deficient mice than from wild-type mice (***Figure 4I***). By immunofluorescence microscopy, Scart2+ γδ T cells were underrepresented in the SCS region in LN sections from Ccr6-deficient

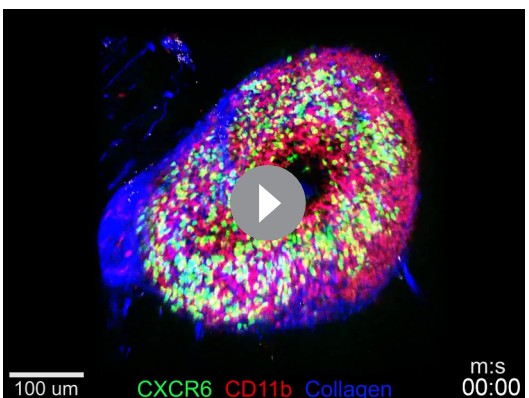

100 um    CXCR6 CD11b Collagen    m:s 00:00

**Video 1.** Cxcr6-GFP$^+$ cell shuttling between parenchyma and the SCS. Representative intravital time-lapse imaging of the popliteal LNs from two Cxcr6$^{GFP/+}$ mice. Overhead 3D video exemplifies the dynamic movement of Cxcr6-GFP$^+$ cells (green) within the LN. Two-dimensional video of a 20 μm maximal intensity projection from an orthogonal plane demonstrates the anatomy of the SCS region. An afferent lymphatic vessel (rarely visualized) drains into the SCS, bounded by the collagenous LN capsule (blue, second harmonic signal) and CD11b$^+$ SCS macrophages (red). The second example further reveals the motility of Cxcr6-GFP$^+$ cells both within the SCS and the LN parenchyma. Cxcr6-GFP$^+$ cells are observed to cross from within the LN parenchyma into the SCS, as well as from within the SCS into the LN parenchyma (examples highlighted by circles). LN, Lymph node; SCS, Subcapsular sinus.

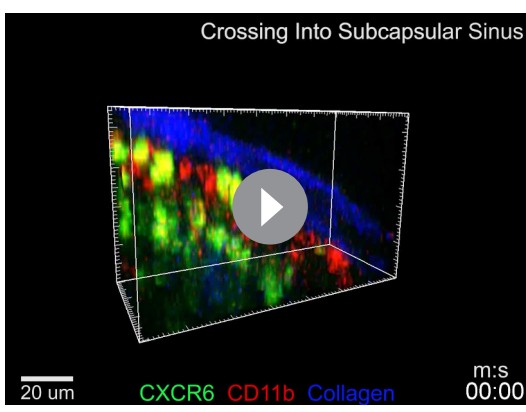

**Video 2.** Representative examples of individual Cxcr6-GFP⁺ cells crossing into and out of SCS. Intravital time-lapse imaging of the popliteal LN from a Cxcr6$^{GFP/+}$ control mouse, highlighting one cell crossing from the parenchyma into the SCS, and one crossing from the SCS into the parenchyma. Cells such as these that clearly crossed from one region were manually identified from automated tracking of all Cxcr6-GFP⁺ cells in an experiment. SCS, Subcapsular sinus.

mice compared with those from control mice (*Figure 4J*, *Figure 4—figure supplement 4*).

Together these data support the conclusion that Ccr6 plays a role in guiding innate-like lymphocytes toward the SCS region. Importantly, when Ccr6 KO mice were immunized with *S. aureus* bioparticles, the IL7Rα$^{hi}$Ccr6⁺ cells mounted a diminished IL17 response (*Figure 4K*), whereas they responded normally to activation stimuli in vitro (not shown). These data provide further evidence that proximity to SCS macrophages is important for innate-like lymphocytes to mount rapid IL17 responses following pathogen exposure.

## S1pr1 is required for innate-like lymphocyte movement into the SCS

S1pr1 is needed in naive lymphocyte for access to cortical and medullary lymphatic sinuses during LN egress (*Cyster and Schwab, 2012*). S1pr1 is also required for marginal zone (MZ) B cell shuttling between the S1P high MZ and the S1P low lymphoid follicle (*Arnon et al., 2013*). We therefore tested whether S1pr1 played a role in guiding innate-like lymphocytes into the SCS. The innate-like lymphocytes had detectable surface S1pr1 (*Figure 5A*) and they responded to S1P by in vitro migration in a Transwell assay (*Figure 5B*). Pretreatment of mice for 6 hr with FTY720, a functional antagonist of S1pr1, greatly diminished in vivo Thy1-PE labeling on IL7Rα$^{hi}$Ccr6⁺ lymphocytes (*Figure 5C*). Similarly, in vivo Scart2 antibody labeling of Vγ4⁺γδ T cells was decreased after FTY720 treatment (*Figure 5D*). FTY720 also decreased the CD169 membrane bleb positive fraction to 20–25% in both the total IL7Rα$^{hi}$Ccr6⁺ population and the γδ T cell subpopulation (*Figure 5C*). The reductions in Thy1-PE-labeled cells (from ~10 to ~1%) and in CD169-bleb⁺ cells (from ~35 to~25%) were similar, representing ~10% of the total IL7Rα$^{hi}$Ccr6⁺ population in both cases (*Figure 5C*), suggesting that the reduced frequency of CD169-bleb⁺ cells was due to the loss of cells accessing the sinus. Comparable findings were made after treatment with the more selective S1pr1 functional antagonist, AUY954 (*Figure 5E*). These data provided evidence that S1pr1 was required for the cells to have normal access to the SCS. Examination of tissue sections by immunofluorescence microscopy showed a loss of Cxcr6$^{GFP/+}$ cells from the SCS following FTY720 treatment (*Figure 5F*). FTY720 treatment also caused a depletion of Scart2⁺ cells from the SCS (*Figure 5G*, *Figure 5—figure supplement 1*). When FTY720-treated mice were challenged with *S. aureus* bioparticles, the IL7Rα$^{hi}$Ccr6⁺ population mounted an IL17 response of normal magnitude (not shown). We speculate that SCS access is needed for other types of responses.

In mice unable to produce lymphatic S1P due to generalized Sphk2 deficiency and ablation of Sphk1 in lymphatic endothelium, innate-like lymphocytes had less CD169⁺ macrophage-derived membrane blebs and less in vivo Thy1-PE antibody labeling compared with control mice (*Figure 5H*). These observations are consistent with the conclusion that the S1P-S1pr1 axis plays a role in guiding innate-like lymphocytes into the SCS.

In accord with S1pr1 having an intrinsic role in promoting SCS access of IL7Rα$^{hi}$Ccr6⁺ cells, staining for S1pr1 and CD169 showed the macrophage-derived bleb coating was restricted to the S1pr1⁺ cells (*Figure 5I*). This analysis required use of an unconjugated rat antibody to detect S1pr1. Since the Thy1-PE is a rat antibody, it was not possible to combine the S1pr1 stain with the in vivo Thy1-PE labeling procedure. To further test whether S1pr1 in innate-like lymphocytes was required for SCS access, S1pr1$^{f/f}$ CreERt2⁺ mice were bred. Tamoxifen treatment of adult mice for 5 days caused a loss of S1pr1 in innate-like lymphocytes (*Figure 5J*). Analysis in separate mice showed that the loss of S1pr1 was associated with reduced in vivo Thy1-PE labeling and reduced CD169 on

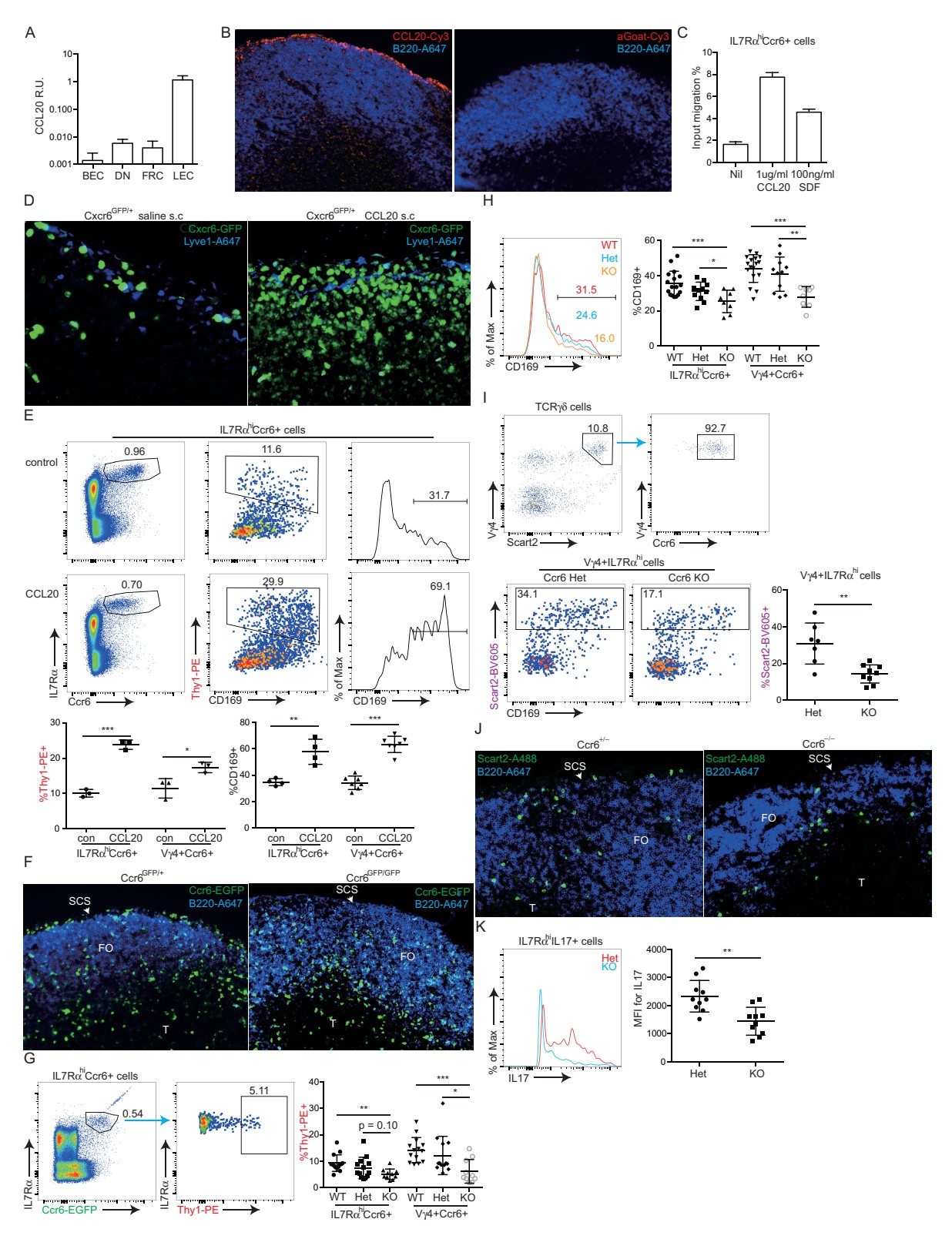

**Figure 4.** Ccr6 promotes innate-like lymphocyte positioning near the SCS. (**A**) *Ccl20* mRNA abundance in sorted LN lymphatic endothelial cells (LEC), blood endothelial cells (BEC), fibroblastic reticular cells (FRC) and double negative stromal cells (DN) determined by qRT-PCR, shown relative to *Hprt*. (**B**) CCL20 staining of LN section (red). The control (no primary) section was stained with the secondary anti-goat-Cy3 antibody alone. B cells were detected in blue (B220). (**C**) Transwell migration of IL7Rα^hi^Ccr6^+^ cells to CCL20. (**D**) Distribution of Cxcr6^GFP/+^ cells in LNs 3 hr after saline or CCL20 s.c.

*Figure 4 continued on next page*

Figure 4 continued

injection. Sections were stained to detect Cxcr6-GFP (green) and Lyve1 (blue). (E) Representative FACS plots show Ccr6 surface level, in vivo Thy1-PE labeling and CD169 macrophage bleb level on IL7Rα$^{hi}$Ccr6$^+$ cells from control (con) or CCL20 injected mice. Summary graph shows comparison of Thy1-PE labeling and CD169+ staining frequency of IL7Rα$^{hi}$Ccr6$^+$ and Vγ4$^+$Ccr6$^+$ cells from control or CCL20 injected mice. (F) LN sections from Ccr6$^{GFP/+}$ or Ccr6$^{GFP/GFP}$ mice stained for EGFP (green) and B220 (blue). White arrow indicates subcapsular sinus area; FO: B cell follicle; T: T zone. (G) Comparison of Thy1-PE labeling on IL7Rα$^{hi}$Ccr6$^+$ cells from WT and Ccr6$^{GFP/+}$ or Ccr6$^{GFP/GFP}$ mice. Ccr6 in Het and KO mice was detected based on GFP reporter expression. (H) Comparison of CD169 staining on IL7Rα$^{hi}$Ccr6$^+$ cells from WT, Ccr6$^{GFP/+}$ or Ccr6$^{GFP/GFP}$ mice. (I) Representative FACS plots showing Vγ4$^+$Scart2$^+$ cells amongst γδT cells and the fraction that are Ccr6+ (upper), and in vivo Scart2-BV605 labeling and CD169 staining (lower). Graph shows summary data. (J) LN sections from Ccr6 Het or KO mice stained for Scart2$^+$ (green) and B220 (blue). White arrow indicates subcapsular sinus area; FO: B cell follicle; T: T zone. (K) Representative histogram plot and summary mean fluorescence intensity (MFI) data of IL17 intracellular staining in IL7Rα$^{hi}$Ccr6$^+$ LN cells from Ccr6 Het or KO mice 3 hr after *S. aureus* bioparticle challenge. *p<0.05, **p<0.01, ***p<0.001, by student's t test. Data are representative of at least two experiments for panel A–D, F, J. Data are representative of two or more experiments with at least two mice per group for panels E, G–I, K. LN, Lymph node; SCS, Subcapsular sinus.

The following figure supplements are available for figure 4:

**Figure supplement 1.** CCL20 distribution in inguinal LN.

**Figure supplement 2.** Movement of Cxcr6-GFP$^+$ cells to SCS location following CCL20 injection.

**Figure supplement 3.** Ccr6 is required for positioning of Ccr6$^+$ cells at the SCS.

**Figure supplement 4.** Ccr6 is required for positioning of Scart2$^+$γδT cells at the SCS.

IL7Rα$^{hi}$Ccr6$^+$ cells (*Figure 5J*). However, this approach could not exclude a role for S1pr1 in other cell types. Attempts to test the intrinsic role of S1pr1 using BM chimeras were unsuccessful due to difficulties in achieving efficient reconstitution of IL7Rα$^{hi}$Ccr6$^+$ cells and our finding that many of the cells developing in the chimeric mice appeared activated based on CD69 expression (not shown). In another approach, mice were treated with tamoxifen for a short time to cause a ~50% reduction in the fraction of cells that were S1pr1$^+$ (*Figure 5K*). We reasoned that if S1pr1 were acting cell intrinsically in IL7Rα$^{hi}$Ccr6$^+$ cells then under conditions of partial ablation the S1pr1-deleted cells should lose their CD169 association whereas this would not occur if the receptor were acting in another cell type. Consistent with an intrinsic role, there was little CD169 staining of S1pr1-negative cells in the tamoxifen-treated mice (*Figure 5K*). These data support the conclusion that S1pr1 acts intrinsically in innate-like lymphocytes to promote close associations with SCS macrophages.

We also examined the effect of S1pr1 antagonism on innate-like lymphocyte migration dynamics using intravital two photon microscopy. Visual inspection of the imaging data for FTY720-treated versus control LNs suggested that there were fewer Cxcr6$^{GFP/+}$ innate-like lymphocytes in the SCS after FTY720 treatment and less examples of cells migrating into the sinus (*Video 3*). Quantification of the number of tracks present in the SCS and the parenchyma in four imaging experiments confirmed that there were fewer cells in the sinus (*Figure 5L*), and there was a significant reduction in the number of tracks that crossed from the parenchyma into the sinus (*Figure 5M*). We also plotted the frequency of cells versus distance from the LN capsule for multiple experiments determined using a computational approach (see Materials and methods) and this confirmed that FTY720 caused a depletion of Cxcr6$^{GFP/+}$ cells from the sinus region (*Figure 5—figure supplement 2*). These data are consistent with the conclusion that S1pr1 antagonism prevents innate-like lymphocyte access to the sinus.

## Involvement of LFA1 and ICAM1 in innate-like lymphocyte access to the LN parenchyma from the SCS

Cell movement from vascular locations into the tissue parenchyma often involves integrin-mediated adhesion. Innate-like lymphocytes express high levels of LFA1 (αLβ2) integrin (*Figure 6A*) and ICAM1 is highly expressed by LECs lining the SCS (*Cohen et al., 2014*) (*Figure 6B*). ICAM1 is also expressed by CD169$^+$ SCS macrophages (not shown). In adhesion assays, IL7Rα$^{hi}$Ccr6$^+$ cells bound avidly to ICAM1 (*Figure 6C*). The Vγ4$^+$ subset had slightly lower LFA1 than the total IL7Rα$^{hi}$Ccr6$^+$ population and adhered less strongly to ICAM1 (*Figure 6A,C*). Given these findings, we

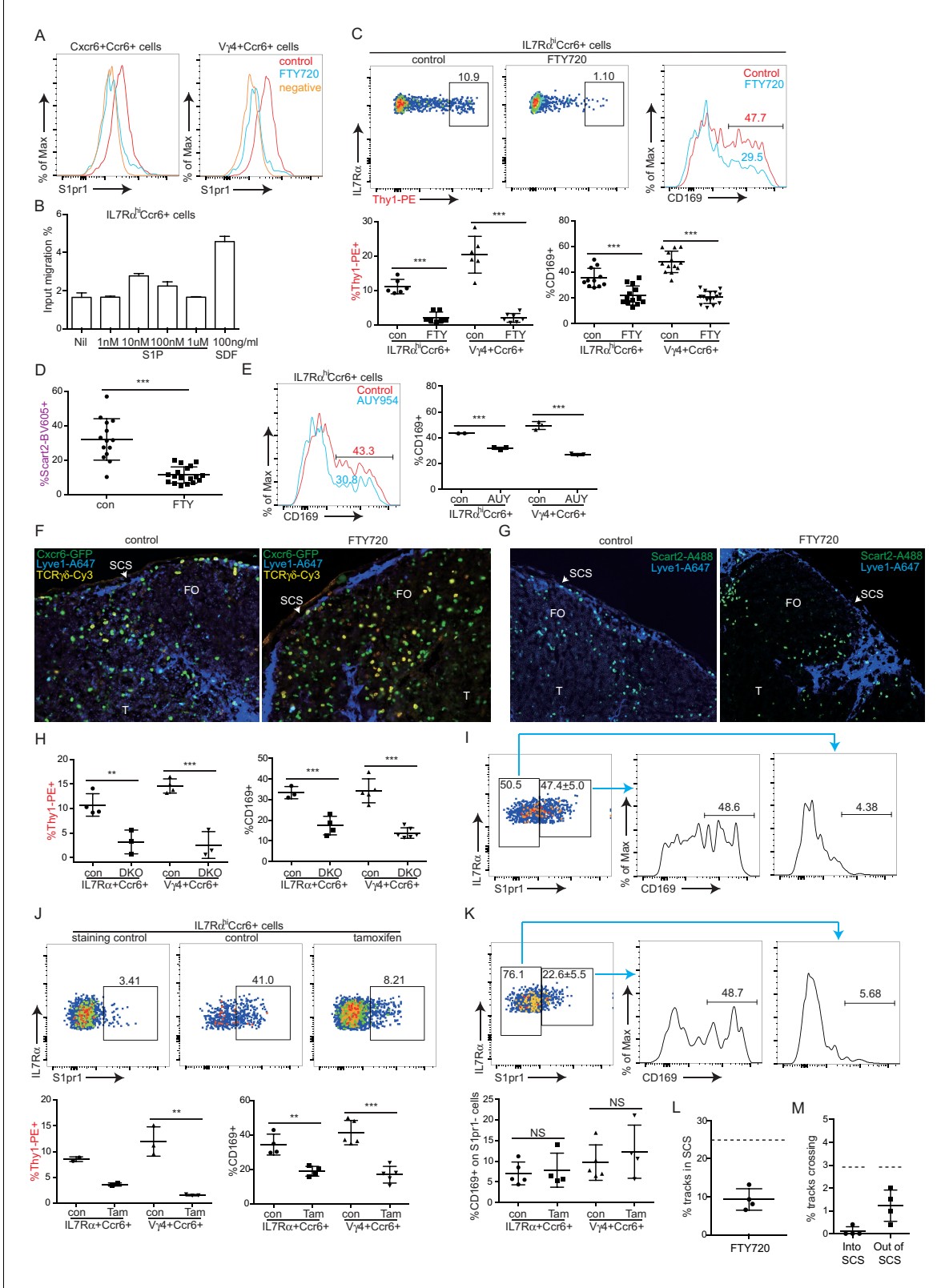

**Figure 5.** S1pr1 is required for innate-like lymphocyte movement into the SCS. (**A**) S1pr1 surface expression on IL7Rα$^{hi}$Ccr6$^+$ and Vγ4$^+$Ccr6$^+$ cells from control or FTY720 treated mice. Negative indicates samples stained with no primary antibody. (**B**) Transwell migration assay showing % of input cells that migrated to the indicated amounts of S1P or SDF. (**C**) Representative FACS plots and summary data of in vivo Thy1-PE labeling and CD169 staining on IL7Rα$^{hi}$Ccr6$^+$ and Vγ4$^+$Ccr6$^+$ LN cells from control or FTY720 treated mice. (**D**) Summary graph showing in vivo Scart2 labeling on

*Figure 5 continued on next page*

Figure 5 continued

IL7Rα$^{hi}$Ccr6$^+$ Vγ4$^+$ cells from control or FTY720-treated mice. (E) Representative histogram plot and summary data of CD169 staining on IL7Rα$^{hi}$Ccr6$^+$ and Vγ4$^+$Ccr6$^+$ LN cells from AUY954-treated mice. (F) GFP, Lyve1 and TCRγδ staining of LN sections from control and FTY720-treated Cxcr6-GFP$^+$ mice. White, SCS: white arrow indicates subcapsular sinus area; FO: B cell follicle; T: T zone. (G) Scart2 and Lyve1 staining of LN sections from FTY720 treated and control mice. White arrow indicates subcapsular sinus area; FO: B cell follicle; T: T zone. (H) Summary data of in vivo Thy1-PE labeling and CD169 staining of the indicated cells in Lyve1-Cre *Sphk1*$^{fl/-}$*Sphk2*$^{-/-}$ (Sphk DKO) and control mice. (I) Representative FACS plot showing S1pr1 staining on IL7Rα$^{hi}$Ccr6$^+$ cells from a control mouse and histogram plots of CD169 staining on the indicated cells. (J) Representative FACS plot showing S1pr1 staining on IL7Rα$^{hi}$Ccr6$^+$ cells from a control and 5-day tamoxifen-treated S1pr1$^{f/f}$ CreERt2 mouse, Graphs show summary data for frequency of Thy1-PE labeled and CD169$^+$ IL7Rα$^{hi}$Ccr6$^+$ and Vγ4$^+$Ccr6$^+$ cells in control (con) or tamoxifen (tam) treated mice. (K) Representative FACS plots of the type in I for cells from a 2-day tamoxifen-treated S1pr1$^{f/f}$ CreERt2 mouse. Graph shows CD169$^+$ cell frequency amongst S1pr1-negative IL7Rα$^{hi}$Ccr6$^+$ cells. *p<0.05, **p<0.01, ***p<0.001, by student's t test. Data are representative of at least two experiments for panels A–B, E–G, I and two or more experiments with at least two mice per group for panels C–D, H. Data are representative of at least three experiments with at least one control and one S1pr1$^{f/f}$ CreERt2 mouse for panels J and K. (L) Percent tracks in SCS compartment among the total tracks enumerated in FTY720 treated mice. Each point represents data from a single movie (three independent experiments). Dashed line is the mean for control mice (data shown in *Figure 3B*). The frequency of cells in the SCS differed significantly from the control (p<0.05 by students t test). (M) Frequency of tracks crossing into and out of the SCS of FTY720-treated mice, enumerated as in L. Dashed lines are the means for control mice (data shown in *Figure 3C*). Frequency of tracks crossing into the SCS differed significantly from the control (p<0.05 by students t test). SCS, Subcapsular sinus.

The following figure supplements are available for figure 5:

**Figure supplement 1.** FTY720 treatment depletes SCART2$^+$γδT cells from the SCS.

**Figure supplement 2.** Frequency of Cxcr6$^{GFP/+}$ cells plotted against their depth from the surface of the LN capsule.

hypothesized that LFA1-ICAM1 interaction may have a role in innate-like lymphocyte movement from the SCS into the LN parenchyma. Consistent with this model, 6-hr αL blocking antibody treatment caused increased Thy1-PE labeling on total IL7Rα$^{hi}$Ccr6$^+$ lymphocytes and on the Vγ4$^+$γδ T cell subset, without influencing their total number in the LN (*Figure 6D*, *Figure 6—figure supplement 1A*). A similar increase in the frequency of Thy1-PE labeled innate-like lymphocytes was observed in *Icam1*$^{-/-}$ mice compared to littermate controls (*Figure 6E*). However, *Icam1*$^{-/-}$ mice had less IL7Rα$^{hi}$Ccr6$^+$ and Vγ4$^+$Ccr6$^+$ cells compared with ICAM1 sufficient control mice, whereas the cells were present at an increased frequency in blood (*Figure 6F,G*). When mice were treated with αL blocking antibody for 3 days, there was a similar reduction in IL7Rα$^{hi}$Ccr6$^+$ cell number in LNs and a marked increase in their numbers in blood (*Figure 6—figure supplement 1B*). These data indicate that, unlike short-term integrin blockade, long-term deficiency of ICAM1 or sustained blockade of LFA1 causes a loss of innate-like lymphocytes from the LN.

By immunofluorescence microscopy, IL7Rα$^{hi}$ cells were found enriched in the SCS in αL blocked mice (*Figure 6H*). These data suggested that innate-like lymphocytes were trapped in the SCS following αL blockade, leading to their increased exposure to the lymph-borne Thy1-PE antibody. To further examine this possibility, intravital two photon microscopy was performed, comparing control and 6 hr αL blocked Cxcr6$^{GFP/+}$ mice. Visual inspection of the movies revealed many Cxcr6$^{GFP/+}$ cells that appeared to be 'stuck' to the SCS floor with their cell bodies 'fluttering' in the sinus, possibly being moved by passing lymph (*Figure 6I* and *Video 4*). These cells had an elongated morphology compared to the rounded shape of cells in the SCS of control

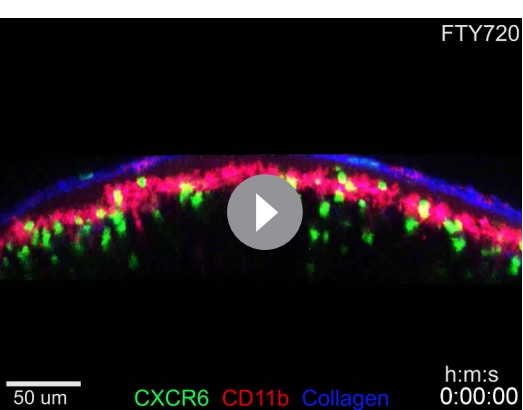

**Video 3.** Cxcr6-GFP$^+$ cellular dynamics following FTY720 treatment. One hour time-lapse imaging of a popliteal LN in a Cxcr6$^{GFP/+}$ mouse 16 hr after treatment with FTY720. Cxcr6-GFP$^+$ cells (green) can be seen accumulated on the parenchymal side of the SCS macrophages (red, CD11b$^+$), and depleted from the SCS. LN capsule appears blue (second harmonic signal).

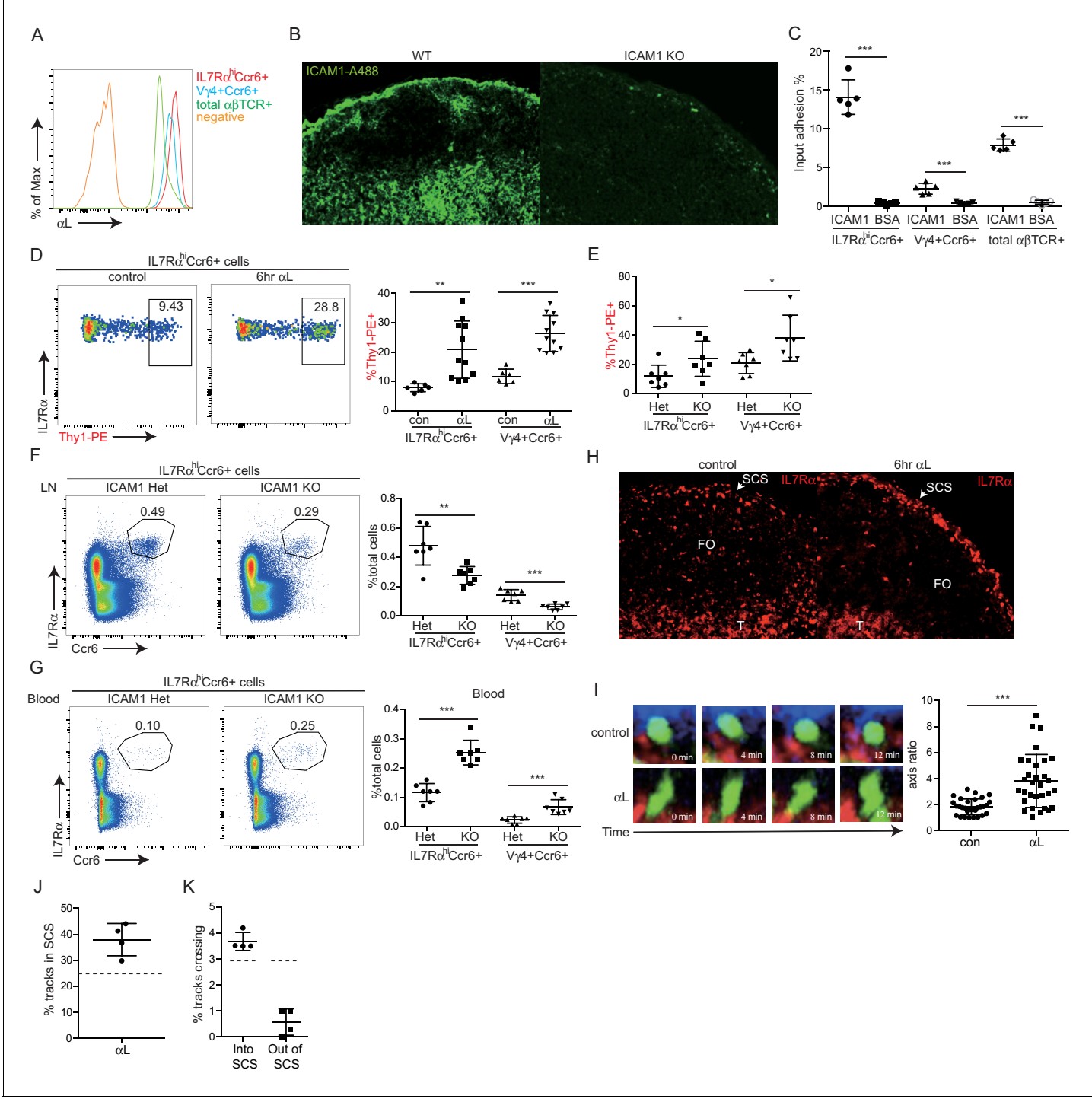

**Figure 6.** LFA1 and ICAM1 control innate-like lymphocyte access to the LN parenchyma from the SCS. (**A**) Representative FACS histogram showing LFA1 staining of IL7Rα$^{hi}$Ccr6$^+$, Vγ4$^+$Ccr6$^+$ and naïve αβ T cells. (**B**) ICAM1 staining of WT and ICAM1 KO LN sections. (**C**) Adhesion of IL7Rα$^{hi}$Ccr6$^+$, Vγ4$^+$Ccr6$^+$ cells and naïve αβ T cells to ICAM1 or BSA. (**D**) Representative FACS plots and summary graph showing in vivo Thy1-PE labeling of IL7Rα$^{hi}$Ccr6$^+$ cells in 6 hr control or αL blocking antibody treated mice. (**E**) Summary graph showing in vivo Thy1-PE labeling of IL7Rα$^{hi}$Ccr6$^+$ cells in ICAM1 Het and KO mice. (**F, G**) IL7Rα$^{hi}$Ccr6$^+$ and Vγ4$^+$Ccr6$^+$ cell frequencies in LNs (**F**) and blood (**G**) from ICAM1 Het and KO mice. (**H**) Distribution of IL7R$^{hi}$ cells in LN sections from control and αL-antibody-treated mice. White, SCS: white arrow indicates subcapsular sinus area; FO: B cell follicle; T: T zone. (**I**) Left panels: Time series of Cxcr6$^{GFP/+}$ cells in LN of control or anti-αL treated mice. Mice were treated with CD11b-PE to label macrophages. Right panel: Axis ratio of Cxcr6$^{GFP/+}$ cells in control and anti-αL treated mice. Cxcr6$^{GFP/+}$ cells in SCS contacting CD11b$^+$ macrophages were measured. Data are pooled from two independent experiments. *p<0.05, **p<0.01, ***p<0.001, by student's t test. Data are representative of at least two experiments for panels **A–C**, **H–I** and two or more experiments with at least two mice per group for panels **D–G**. (**J**) Percent of tracks in the SCS

*Figure 6 continued on next page*

Figure 6 continued
compartment among the total tracks enumerated in anti-αL-treated mice. Each point represents data from a single movie (three independent experiments). Dashed line is the mean for control mice (data shown in *Figure 3B*). The frequency of cells in the SCS differed significantly from the control (p<0.05 by students t test). (**K**) Frequency of tracks crossing into and out of the SCS of anti-αL-treated mice, enumerated as in J. Dashed lines are the means for control mice (data shown in *Figure 3C*). Frequency of tracks crossing out of the SCS differed significantly from the control (p<0.05 by students t test). SCS, Subcapsular sinus.
The following figure supplement is available for figure 6:
**Figure supplement 1.** Effects of αL blockade on innate-like lymphocyte distribution.

mice (*Figure 6I* and *Video 4*). Consistent with the in vivo Thy1-PE labeling and IF microscopy, quantitative analysis of the imaging experiments by enumerating cell tracks and by a computational approach revealed that αL treatment caused an increase of cells in the sinus in association with the macrophage layer (*Figure 6J*, *Figure 6—figure supplement 1C*). Moreover, the cell tracking analysis showed that while there was not a significant change in the frequency of tracks crossing from the parenchyma into the SCS, there was a significant reduction in cells traveling from the SCS into the parenchyma (*Figure 6K*). These data support the conclusion that LFA1-ICAM1-mediated adhesion is required for innate-like lymphocytes to migrate from the SCS into the LN parenchyma.

## CD169 mediates SCS retention of innate-like lymphocytes

Our finding that innate-like lymphocytes are coated with CD169[+] macrophage-derived membrane blebs (*Gray et al., 2012*) together with in vitro evidence that CD169 can support adhesion of certain cell types (*Crocker et al., 1995*; *Crocker and Gordon, 1989*; *van den Berg et al., 2001*) led us to test whether CD169 contributed to innate-like lymphocyte migration or adhesion in the SCS. CD169 binds α2–3 linked sialic acid on surface glycoproteins (*Macauley et al., 2014*). By CD169-Fc staining, we found IL7Rα[hi]Ccr6[+] lymphocytes had a high amount of CD169 ligand on their surface, and neuraminidase treatment of the cells abolished their ability to bind CD169 (*Figure 7A*). The acquisition of abundant macrophage membrane fragments by innate-like lymphocytes was dependent on CD169, since in mice in which CD169 was blocked with a neutralizing antibody (*Crocker and Gordon, 1989*) or in mice deficient in CD169, there were no CD11b[+] macrophage membrane fragments on innate-like lymphocytes (*Figure 7B* and *Figure 7—figure supplement 1A*).

By real-time two photon microscopy, CD169 blockade in Cxcr6[GFP/+] mice appeared to increase the amount of innate-like lymphocyte migration within the SCS and between parenchyma and sinus (*Video 5*). Quantitation of the cell tracks crossing between compartments confirmed that there was an increase in bidirectional exchange (*Figure 7C*). We also examined Cxcr6-GFP[+] cells in CD169 KO mice and here too the cells seemed to move extensively in the SCS region (*Video 5*). These data are consistent with the idea that CD169 blockade or deficiency disrupts stable interactions between innate-like lymphocytes and SCS macrophages.

We therefore asked whether CD169 plays a role in mediating lymphatic sinus retention of innate-like lymphocytes. After 30 hr of treatment with CD169-blocking antibody, there was no change in total IL7Rα[hi]Ccr6[+] cell frequency, but

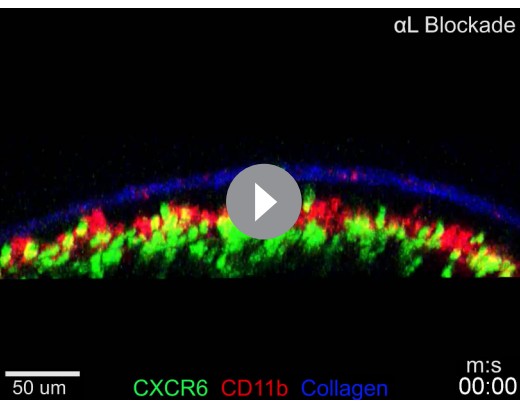

**Video 4.** Cxcr6-GFP[+] cell fluttering in the SCS following αL blockade. In Cxcr6[GFP/+] mice, four hours after treatment with αL blocking antibody, Cxcr6-GFP[+] cells (green) are observed to flutter at the floor of the SCS. Cxcr6-GFP[+] cells appear attached to CD11b[+] SCS macrophages (red) while being buffeted by bulk lymph flow in the SCS, yielding a characteristic fluttering dynamic (video inset and arrowheads). LN capsule appears blue. LN, Lymph node; SCS, Subcapsular sinus.

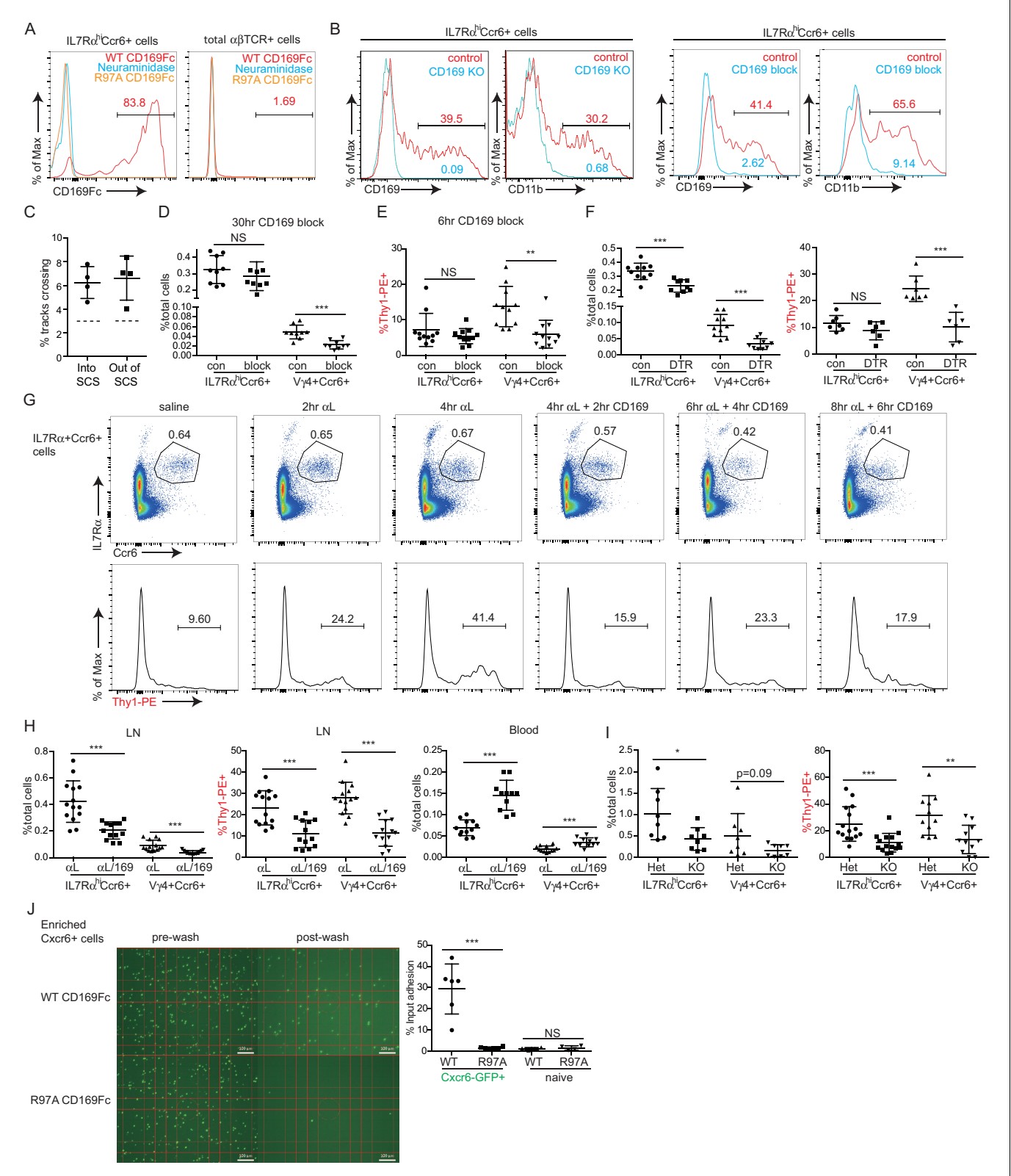

**Figure 7.** CD169 mediates SCS retention of innate-like lymphocytes. (**A**) CD169-Fc binding of innate-like lymphocytes. (**B**) Effect of CD169-deficiency or blocking antibody treatment on macrophage bleb acquisition by IL7Rα^hiCcr6+ cells. Cells were stained to detect the macrophage markers CD169 and CD11b. (**C**) Frequency of tracks crossing into and out of the SCS of anti-CD169-treated mice. Each point represents data from a single movie (three independent experiments). Frequency of tracks crossing into and out of the SCS differed significantly from the control (p<0.05 by students t test). (**D**)
*Figure 7 continued on next page*

Figure 7 continued

Number of Vγ4⁺Ccr6⁺ cells in LN after 30 hr of CD169 blockade. (E) In vivo Thy1-PE labeling on Vγ4⁺Ccr6⁺ cells after 6 hr CD169 blockade. (F) Number and in vivo Thy1-PE⁺-labeled Vγ4⁺Ccr6⁺ cell frequency in LNs of control (con) or CD169-DTR⁺ mice after DT treatment. (G) Change in innate-like lymphocyte frequency and Thy1-PE labeling over time after treating mice with αL and CD169 blocking antibodies. (H) Effect of 6 hr combined αL and CD169 blockade on IL7Rα^hiCcr6⁺ cell frequency in LN and blood, and fraction of LN cells that are in vivo Thy1-PE labeled. (I) Effect of αL blocking in CD169 KO mice on IL7Rα^hiCcr6⁺ cell number and in vivo Thy1-PE labeling. (J) IL7Rα^hiCcr6⁺ cell adhesion to CD169-Fc and R97A-Fc-coated plates. *p<0.05, **p<0.01, ***p<0.001, by student's t test. Data are representative of at least two experiments for panels A–B, F, I. Data are representative of two or more experiments with at least two mice per group for panels C–E, G–H. LN, Lymph node; SCS, Subcapsular sinus.

The following figure supplement is available for figure 7:

**Figure supplement 1.** Effects of CD169 blockade on innate-like lymphocyte properties and distribution.

there was a two-fold loss of Vγ4⁺Ccr6⁺ cells in the LN (**Figure 7D**). We speculated that if this loss was occurring due to inhibited adherence of cells in the SCS then the lymph exposed cells might be lost rapidly following CD169-blockade. Indeed, in 6-hr blockade experiments, there was a significant reduction in the fraction of Vγ4⁺Ccr6⁺ cells that were Thy1-PE labeled (**Figure 7E**). There was only a slight reduction in the total numbers of Vγ4⁺Ccr6⁺ cells after this short period (**Figure 7—figure supplement 1B**) consistent with only a low fraction of the total population being in the sinus at a given time. These observations suggest that CD169 blockade caused a loss of cells in lymphatic sinuses and this in turn led to a loss of cells from the LN over time. A loss of Vγ4⁺Ccr6⁺ cells was also observed following CD169⁺ macrophage ablation using CD169-DTR mice (**Figure 7F**). In considering explanations for why these effects were mostly selective to the Vγ4⁺Ccr6⁺ subpopulation, we found it notable that in ICAM1 adhesion assays, Vγ4⁺Ccr6⁺ cells were less capable of binding ICAM1 compared with other innate-like lymphocytes (**Figure 6C**). Taking this observation together with the finding that a higher fraction of Vγ4⁺Ccr6⁺ cells were lymph-exposed in the steady state (**Figure 3D**), we speculated that the non-γδT innate-like lymphocytes – but not the Vγ4⁺Ccr6⁺ cells – in the SCS were able to travel back into the LN parenchyma even in the absence of CD169.

In an effort to reveal a role of CD169 in sinus retention of the total innate-like lymphocyte population, we blocked αL prior to blocking CD169. In the first 4 hr following αL treatment, we observed a gradual increase in the fraction of cells that were Thy1-PE⁺ (**Figure 7G**), consistent with the accumulation of cells in the sinus (**Figure 6**). Within 2 hr of anti-CD169 treatment, this enhanced Thy1 labeling was lost, indicating loss of innate-like lymphocytes from the sinus, and there was a reduction in IL7Rα^hiCcr6⁺ cell frequency (**Figure 7G**). IL7Rα^hiCcr6⁺ cell frequency declined further after 4 and 6 hr of double blockade (**Figure 7G**). When αL and CD169 were both blocked continually for 6 hr, there was a 50% loss of total innate-like lymphocytes in LNs, and this was accompanied by an increase of the cells in blood (**Figure 7H**). In CD169 Het and KO mice, there were comparable starting frequencies of IL7Rα^hiCcr6⁺ cells (**Figure 7—figure supplement 1C**), but after αL blocking antibody treatment there was a loss of Thy1-PE labeling and in total IL7Rα^hiCcr6⁺ cells in the CD169 KO mice compared with the Het controls (**Figure 7I**).

The decrease in innate-like lymphocytes in LNs 6 hr after αL and CD169 double blockade was also evident in intact LNs visualized by real-time

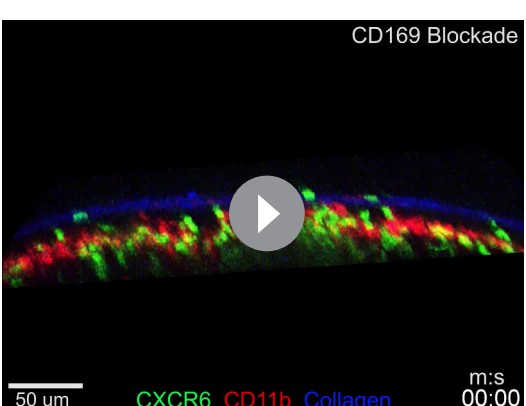

**Video 5.** Increased movement of SCS Cxcr6-GFP⁺ cells after CD169 blockade and in CD169⁻/⁻ mice. Representative time-lapse images of Cxcr6^GFP/+ mice 4 hr after treatment with CD169 blocking antibody, as well as Cxcr6^GFP/+ CD169⁻/⁻ mice. In both conditions, there appeared to be an increased frequency of Cxcr6-GFP⁺ cell (green) crossing events, both from the SCS into the LN parenchyma and from the parenchyma into the SCS. LN capsule appears blue and SCS macrophages (CD11b) red. LN, Lymph node; SCS, Subcapsular sinus.

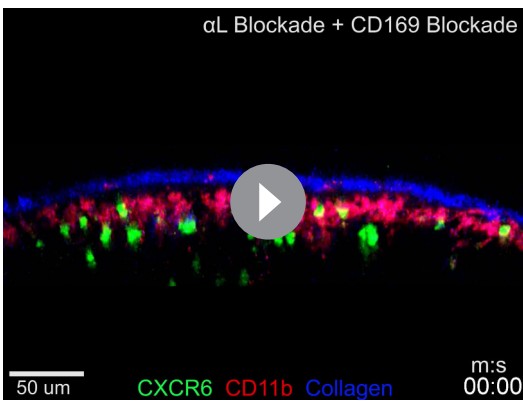

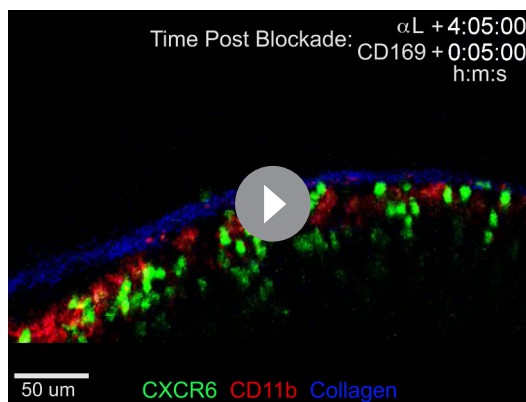

**Video 6.** Decreased Cxcr6-GFP$^+$ cell frequency in SCS following αL and CD169 double blockade. Decreased numbers of Cxcr6-GFP$^+$ cells (green) are observed in the SCS and LN parenchyma following dual antibody blockade of αL and CD169. Time-lapse imaging in a Cxcr6$^{GFP/+}$ mouse beginning 4 hr post treatment. LN capsule appears blue and SCS macrophages (CD11b) red. LN, Lymph node; SCS, Subcapsular sinus.

**Video 7.** Addition of CD169 blockade causes release of Cxcr6-GFP$^+$ cells from floor of SCS when pre-treated with αL blocking antibody. Representative time-lapse imaging beginning 5 min after treatment with CD169 blocking antibody in Cxcr6$^{GFP/+}$ mice pretreated 4 hr before with αL blocking antibody. Upon addition of CD169 blockade, Cxcr6-GFP$^+$ cells (green) detached from the floor of the SCS and entered the bulk lymph flow, rapidly moving away from the field of view (inset and arrowheads). LN capsule appears blue and SCS macrophages (CD11b) red. LN, Lymph node; SCS, Subcapsular sinus.

two photon microscopy (*Video 6*). Importantly, unlike αL blockade, which caused many innate-like lymphocytes to become non-migratory and apparently stuck to the SCS floor (*Video 4*), applying anti-CD169 in addition to anti-αL led to innate-like lymphocyte detachment from CD169$^+$ macrophages and loss in the lymph flow (*Video 7*). In some regions, it was also possible to observe cells moving from the parenchyma into the sinus, but then failing to attach and being carried away in the lymph flow (*Video 7*). By quantitative analysis of three imaging experiments, CD169-blockade or deficiency was found to have little effect on the density of Cxcr6$^{GFP/+}$ cells in the SCS, but combined αL and anti-CD169 treatment caused a decrease of cells from the sinus region (*Figure 7—figure supplement 1D,E*). These data provide evidence that CD169 has a role in mediating lymphatic sinus retention of most innate-like lymphocytes.

Finally, we examined the sufficiency of CD169 to support adhesive interactions of innate-like lymphocytes. In adhesion assays, IL7Rα$^{hi}$Ccr6$^+$ lymphocytes showed binding to plates coated with recombinant WT but not a binding site mutant of CD169 (*Figure 7J*). By contrast, naïve αβ T cells showed minimal binding to CD169 (*Figure 7J*).

## Discussion

The above findings show that the Ccr6-dependent positioning of IL7Rα$^{hi}$Ccr6$^+$ innate-like lymphocytes near the LN SCS is important for their rapid cytokine production following bacterial or fungal challenge. The data support a model (*Figure 8*) where lymphatic endothelial-derived CCL20 acts on Ccr6 to attract the innate-like lymphocytes into proximity with SCS macrophages. This enhances their exposure to macrophage-derived cytokines that promote IL17-production and likely other effector functions of the lymphocytes

Our findings also reveal an unusual migratory behavior of innate-like lymphocytes in the SCS region that involves exchange of cells between the parenchyma and SCS. Movement across the CD169$^+$ macrophage layer into the sinus is promoted by S1pr1 and lymphatic endothelial cell-derived S1P. Within the sinus CD169 on macrophages binds to sialylated ligands on the innate-like lymphocytes and helps prevent loss of the cells in lymph flow, with the Vγ4+ subset of innate-like lymphocytes being most dependent on this adhesive system. LFA1 binding to ICAM1 also

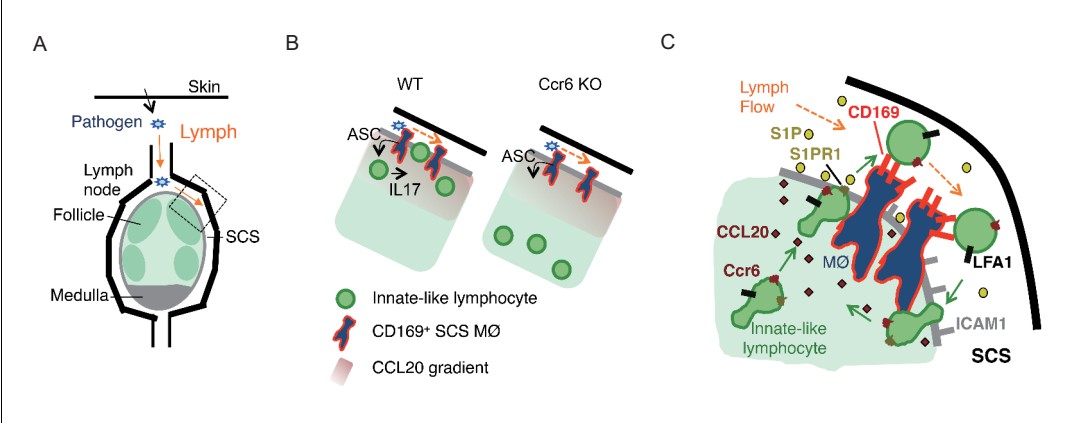

**Figure 8.** Model of requirements for innate-like lymphocyte surveillance of the LN SCS. (**A**) Diagram of skin draining LN. (**B**) Model showing effect of Ccr6-deficiency on innate-like lymphocyte positioning and associated defect in ability to upregulate IL17 in response to IL1-family cytokines produced by SCS macrophages in an ASC-dependent manner. (**C**) Model showing role of Ccr6-CCL20 in guiding innate-like lymphocyte to lymphatic sinus, S1pr1-S1P in promoting trans-cellular migration into sinus, CD169 on macrophage (MØ) in mediating retention of CD169-ligand$^{hi}$ lymphocyte (green) against lymph flow, and LFA1-ICAM1 in promoting adhesion and transmigration. Green arrows show cell migration and orange arrows show lymph flow. LN, Lymph node; SCS, Subcapsular sinus.

contributes to retaining innate-like lymphocytes in the sinus. Finally, return of cells to the parenchyma involves LFA1 and ICAM1, presumably to support transmigration across the lymphatic endothelium. Although this local migratory behavior was not essential for mounting IL17 responses against the pathogens tested in this study (data not shown), we propose that this surveillance program allows innate like lymphocytes to interrogate the CD169$^+$ macrophages and SCS for pathogen- or commensal-derived molecules for which they have appropriate receptors.

The cues required for homeostatic positioning of cells in LN T zone and follicles have been well studied, with CCL21/CCL19 and CXCL13 having dominant roles (*Cyster, 2005*). Our findings add CCL20 as an additional homeostatic organizer, acting to recruit Ccr6$^+$ IL17-committed cells to the LN SCS region. As well as expression by innate-like lymphocytes, Ccr6 is abundant on Th17 cells (*Littman and Rudensky, 2010*) and we speculate that Th17 effector cells that remain in the LN following immunization or infection may position in proximity with the SCS. Although Cxcr6 is also abundantly expressed by innate-like lymphocytes, Cxcr6-deficiency did not appear to affect their distribution in the LN (unpubl. obs). The expression of CCL20 in a subcapsular region in primate LNs (*Choi et al., 2003*; *Pegu et al., 2007*) makes it likely that our findings with Ccr6 in mice will extend to humans. Ccr6 antagonism was suggested to reduce the egress of Ccr6$^+$ effector CD4 T cells from LNs in a mouse EAE model (*Liston et al., 2009*). CCL20 can be strongly induced in inflamed tissue (*Mabuchi et al., 2013*), and we speculate that under the inflammatory condition associated with EAE, CCL20 travels to LNs from the inflamed site and attracts Ccr6$^+$ cells into the sinus lumen as we observed following CCL20 injection. In the case of Ccr6$^+$ CD4 effector T cells, this might then favor their exit from the LN.

S1P and S1pr1 have a critical role in promoting egress of T and B cells from LNs, acting at the step of transmigration into cortical and medullary sinuses (*Grigorova et al., 2009*; *Sinha et al., 2009*). We describe here a further function for this ligand-receptor pair in promoting cell migration into the SCS. As for movement into cortical and medullary sinuses, this response involves S1P production by lymphatic endothelial cells. S1pr1 on naive lymphocytes is required during the egress commitment step and did not appear to have a role in promoting chemotaxis to the sinus (*Grigorova et al., 2009*). Consistent with those findings, we did not observe an obvious effect of S1pr1 antagonism on innate-like lymphocyte density near the SCS, only a loss of cells from within the sinus. We therefore favor the model that the cells approach the sinus-lining lymphatic endothelial cells in a CCL20-dependent manner and S1pr1 commits some of the cells to cross the lymphatic endothelium (and associated CD169$^+$ macrophage layer) and enter the sinus. Within the sinus, we anticipate that the cells are exposed to high amounts of S1P that cause rapid, GRK2-dependent

(*Arnon et al., 2011*), internalization and desensitization of S1pr1, allowing the cells to respond to cues (possibly including CCL20) that can promote their return to the parenchyma. This type of shuttling behavior has been described in the spleen for another population of innate-like lymphocytes, the MZ B cells, with movement into the blood-rich MZ being S1pr1 dependent and return to the parenchyma being CXCL13 dependent (*Arnon et al., 2013*). One function of MZ B cell shuttling is to deliver immune complexes from blood to B cell follicles (*Cinamon et al., 2008*). It will be interesting to see if cell shuttling in LNs contributes to cargo delivery from lymph to the LN parenchyma.

CD169 is abundantly expressed on the lymph-exposed heads of SCS macrophages as well as on the tails that extend into the LN parenchyma (*Phan et al., 2007*). We show that CD169 supports adhesion of innate-like lymphocytes in the sinus and helps restrain them against lymph flow. A number of in vitro studies have shown an ability of CD169 to support cell-cell adhesion (*Crocker et al., 1995*; *Crocker and Gordon, 1989*; *van den Berg et al., 2001*). Recent work has also revealed that CD169 on LN SCS macrophages can play a role in the capture of lymph-borne exosomes (*Saunderson et al., 2014*) and retroviruses (*Sewald et al., 2015*). The present work builds upon those observations to provide in vivo evidence that this Siglec family member can support shear stress-resistant adhesion of cells. Although lymph flow rates in the mouse popliteal LN have not been directly measured, a modeling study estimated wall shear stresses of several dyn/cm$^2$ in the SCS close to afferent lymphatic vessels (*Jafarnejad et al., 2015*). These shear stresses are similar to those in blood vessels that support leukocyte adhesion (*Finger et al., 1996*). A recent genomics study found that high endothelial venules in Peyer's patches highly express St6gal1, an enzyme that can generate ligands for CD22 (Siglec-2) (*Lee et al., 2014*). In transfer experiments, CD22-deficient B cells showed reduced homing to Peyer's patches (*Lee et al., 2014*). CD22 also contributes to B cell homing to the bone marrow (*Nitschke et al., 1999*). These studies together with our work provide evidence that Siglecs are a second class of lectin, after the selectins, that functions in mediating shear-resistant adhesion of cells. A key feature that facilitates selectin function is the high density of glycosylated ligands on the target cell and the very rapid on- and off-rates of lectin-ligand interactions (*Rosen, 2004*). We suggest that similar features contribute to the function of CD169 as a shear-resistant adhesion receptor. The long ectodomain of CD169 (with 17 Ig-domains) also seems likely to contribute to an ability to 'capture' cells that have become dislodged by lymph flow and to allow their re-adhesion and subsequent transmigration. A similar activity may be involved in capturing CD169-ligand high cells arriving in the SCS from the afferent lymphatic.

Our study provides evidence that LFA1 and ICAM1 are required for retaining IL7Rα[hi]Ccr6[+] innate-like lymphocytes in the SCS and supporting their return from the sinus to the parenchyma. These findings are in accord with the well-established roles of LFA1 and ICAM1 in supporting adhesion to and transmigration of lymphocytes across blood vessel endothelium into tissues (*Rot and von Andrian, 2004*). However, they are in discord with a study showing that migration of BM-derived DCs from the SCS into the LN parenchyma was integrin independent (*Lämmermann et al., 2008*). The basis for this discrepancy is not yet clear but might reflect general differences in the properties of lymphocytes and BM-derived DCs. Moreover, integrins may contribute to DC trafficking into LNs under conditions of inflammation (*Teijeira et al., 2013*). Our conclusion that LFA1 and ICAM1 function during innate-like lymphocyte movement from the SCS into the LN parenchyma is based on four sets of observations in mice treated with αL blocking antibody: (1) innate-like lymphocytes transiently accumulate in the SCS; (2) the accumulated cells exhibit an unusual elongated morphology (suggesting a less adhesive state); (3) tracking analysis shows reduced numbers of cells migrating from the SCS into the parenchyma; (4) the cells are more sensitive to dislodgement by CD169 blocking antibody. We feel that the most parsimonious explanation for these data is that LFA1 supports adhesive interactions between innate-like lymphocytes and ICAM1$^+$ sinus-associated cells (lymphatic endothelial cells, macrophages) that are needed for movement into the parenchyma. However, given the heterogeneity of the innate-like lymphocyte population, we cannot exclude the possibility that LFA1 blockade also decreases retention of some cells within the LN parenchyma, thereby increasing their movement into the SCS. A study of naive lymphocyte egress from LNs showed that LFA1 and ICAM1 can contribute to promoting retention of cells in the LN and this was suggested to reflect a role for LFA1 in limiting the rate of cell movement across the endothelium into the sinus lumen (*Reichardt et al., 2013*). More studies will be needed to fully address all the functions of LFA1 and ICAM1 in innate-like lymphocyte migration dynamics.

The migration of innate-like lymphocytes in close association with SCS macrophages seems likely to ensure that factors made by the macrophages, such as IL1-family cytokines, can rapidly engage the lymphocytes. It presumably also ensures that signaling in the reverse direction, from the innate-like lymphocyte to the CD169[+] cells, can take place efficiently. Such signaling may serve to augment the antibacterial, antifungal or antiviral activities of the macrophages. The importance of cell movement into the lumen of the SCS is not yet clear but might allow prompt surveillance of pathogens and endogenous cells (e.g. cancer cells) arriving via the lymph, prior to their accessing the LN parenchyma. The greater CD169 coating of Vγ4[+]γδT cells and their stronger dependence on CD169 for retention in the SCS than the other innate-like lymphocytes suggests these cells interact more strongly or in a more selective way with CD169[+] macrophages. This likely reflects requirements for recognizing and responding to unique ligands beyond IL1β and IL23.

Following *Toxoplasma* infection or after vaccinia virus injection, NK cell interaction with CD169[+] macrophages is induced (*Coombes et al., 2012*; *Garcia et al., 2012*). These studies observed NK cell movement on collagen fibers and of NK cells slowing or stopping in contact with CD169[+] cells. Whether collagen fibers guide the movement of IL7Rα[hi]Ccr6[+] cells needs further study, although in contrast to NK cells, the IL7Rα[hi]Ccr6[+] cells had minimal expression of the collagen binding α2 integrin (not shown and [*Gray et al., 2012*]). Another distinction between these cell types is that LN NK cells lack Ccr6 (unpubl. obs). A previous study showed that NK1.1[+] cells in the LN are concentrated in medullary regions (*Kastenmüller et al., 2012*), consistent with their homeostatic positioning being controlled by cues other than CCL20. NK cell movement to the SCS might occur in response to inflammation-induced chemoattractants such as CXCR3 ligands that can be upregulated in this region and function in recruiting activated CD4 T cells and CD8 central memory cells (*Garcia et al., 2012*; *Groom et al., 2012*; *Sung et al., 2012*). NKT cell interaction with SCS macrophages was observed following immunization with α-galactosylceramide (*Barral et al., 2010*). Since 15–20% of the IL7Rα[hi]Ccr6[+] innate-like lymphocytes are NKT cells (*Gray et al., 2012*), it is likely that these cells are continually surveying the SCS macrophages in a manner similar to the bulk Cxcr6[+] population studied here. As well as macrophages, there are DCs in the SCS region (*Gerner et al., 2015*) and the migration behavior we describe may help ensure efficient surveillance of SCS-associated DCs. Following exposure to strong inflammatory signals, SCS macrophages move into the follicle (*Gaya et al., 2015*). It will be interesting to examine how this reorganization modifies the innate-like lymphocyte trafficking behavior.

Previous work has shown that the IL7Rα[hi]Ccr6[+]γδT cells in peripheral LNs and related cells in the dermis are precommitted to IL17 production, with stimulation by IL1β and IL23 being sufficient to strongly promote IL17 production by these cells (*Cai et al., 2011*; *Gray et al., 2011*; *Haas et al., 2009*; *O'Brien and Born, 2015*; *Ramirez-Valle et al., 2015*). Our studies here show that the IL7Rα[hi]Ccr6[+]αβT cell population also readily produces IL17 upon IL1β and IL23 stimulation. These cells are double negative for CD4 and CD8 (*Gray et al., 2012*). LN DN T cells highly express IL23R and respond to this cytokine (*Mizui et al., 2014*; *Riol-Blanco et al., 2010*). Our findings in ASC-deficient mice provide in vivo evidence that IL1-family cytokines are involved in activating the cells. These observations are reminiscent of findings for innate-like CD8 T cells in LNs that rapidly make IFNγ upon cytokine (IL18 and IL12 or IL18 and IFNα) stimulation, and for various types of memory T cells that make IFNγ upon IL12 and IL18 exposure (*Jameson et al., 2015*; *Kastenmüller et al., 2012*). While our data suggest cytokines may be sufficient to activate the innate-like T cells under some conditions, this does not exclude a role for TCR stimulation in promoting activation or augmenting responses under other conditions.

In summary, we demonstrate that IL17-committed innate-like lymphocytes survey the pathogen-exposed surface of peripheral LNs and respond rapidly upon challenge with bacterial and fungal pathogens. Ccr6-guided proximity to the SCS is important for these cytokine-driven responses. S1pr1, CD169 and LFA1 function to promote migration between parenchyma and SCS in close association with CD169[+] macrophages in a program that we suggest allows innate-like lymphocytes to survey for a range of pathogen- and commensal-derived antigens and mount appropriately tailored responses.

## Materials and methods

### Mice

Wild-type C57BL/6NCr mice of 7–9 weeks of age were purchased from the National Cancer Institute (Frederick, MD). $Cxcr6^{GFP/+}$ (RRID:MGI:3616633), $Ccr6^{GFP/+}$ (RRID:MGI:3852186), $Ccr6^{+/-}$ (RRID: MGI:4359785), $Pycard^{+/-}$, CAG-KiKGR$^+$, $S1pr1^{f/f}$, CreERt2$^+$, Lyve1-Cre$^+$Sphk1$^{f/f}$ Sphk2 $^{-/-}$ mice (RRID:MGI:4421697), and Icam1$^{+/-}$ mice were previously described and were from JAX or from an internal colony. CD169-DTR mice were provided by Masato Tanaka (*Miyake et al., 2007*). Littermate mice were evenly distributed into control or treatment groups and mice of both groups were co-caged whenever possible. All mice were adult and were studied between 7 and 20 weeks of age. Animals were housed in a specific-pathogen-free environment in the Laboratory Animal Research Center at the University of California, San Francisco, and all experiments conformed to ethical principles and guidelines approved by the Institutional Animal Care and Use Committee, protocol approval number: AN107975-02.

### FTY720 and AUY954 treatment

Mice were treated with FTY720 in saline at a dose of ~1 ug/g i.p. Mice were analyzed for Thy1-PE labeling or CD169 bleb association at 6 hr or O/N post treatment. Control mice were treated with saline i.p. AUY954 in saline was given i.p. in 300 µl at 300 µM. Mice were analyzed for CD169 bleb association O/N post treatment. Control mice were treated with saline i.p.

### αL and anti-CD169 blocking

100 µg αL blocking antibody (clone M17/4) or anti-CD169 blocking antibody (Ser4) or both antibodies was injected i.v. Mice were analyzed 6 hr or 30 hr post treatment. For 3-day experiments, mice were injected with 100 µg αL blocking antibody at day-3 and day-1, and experiments were done at day 0.

### Footpad bacterial challenge

Mice were challenged with $2 \times 10^7$ CFU of heat inactivated *C. albicans*, attenuated *Yersinia pestis*, or 150 µg *S. aureus* bioparticle (Invitrogen, Cat S2859) through the footpad. Control mice were treated with 25 µl saline. Draining popliteal LNs were dissected and analyzed 3 hr post challenge. For IL17 staining, popliteal LN cells were incubated in Golgi plug for 2 hr at 37°C, stained for surface antigens, treated with BD Cytofix Buffer and Perm/Wash reagent (BD Biosciences), and stained with anti-IL-17A.

### In vivo Thy1-PE labeling

0.2 µg Thy1.2-PE antibody (30-H12) was injected through the footpad in 25 µl saline. Labeling was done for 5 min. Draining popliteal LNs were harvested for flow cytometric or immunofluorescence analysis.

### Diphtheria toxin treatment

WT and CD169-DTR/+ were treated with 0.75 µg DT on Day-5 and Day-2. Experiments were performed and the mice analyzed on Day 0.

### Tamoxifen treatment

For full deletion, WT and $S1pr1^{f/-}$ ERcre$^+$ mice were treated with tamoxifen at Day -5 to Day -1 and analyzed on Day 0. For transient deletion, mice were treated with one dose on Day -2 and analyzed on Day 0. Tamoxifen was dosed at 5 mg/mouse/day orally.

### Flow cytometry

Cells were stained in 'FACS buffer' (PBS with 0.1% sodium azide, 2% FBS and 1 µM EDTA) with antibodies to TCRγδ (GL3), TCRβ (H57-597), IL17A (eBio17B7), Ccr6 (140706), IL7Rα (A7R34), CD3ε (clone 145-2C11), CD11b (clone Mac-1), Vγ4 (clone UC3-10A6), CD90.2 (30-H12), S1pr1 (R&D, MAB7089, clone 713412), anti-CD169 (clone MOMA-1 and clone Ser4); anti-scart2 antibody was kindly provided by Dr. Klaus Karjalainen. Molecular Probes Monoclonal Antibody Labeling Kits

(Invitrogen) were used to directly conjugate antibody to Alexafluor647 or Pacific Blue dyes. During analysis, singlets were gated based on peak FSC-H/FSC-W and SSC-H/SSC-W. These gates encompassed more than 90% of total events and were set sufficiently wide to include singlet events of variable size while avoiding the main doublet peak.

To detect IL-17A, cells were stimulated for 3 hr with 50 ng/ml phorbol 12-myristate 13-acetate (PMA, Sigma) and 1 µg/ml Ionomycin (I, EMD Biosciences) or 3 hr with 10 ng/ml IL1β and 10 ng/ml IL23 in Golgi plug (BD Biosciences) at 37°C, stained for surface antigens, treated with BD Cytofix Buffer and Perm/Wash reagent (BD Biosciences), and stained with anti-IL-17A.

## CD169-Fc/R97A CD169-Fc FACS staining

1 µg/ml CD169-Fc or R97A CD169-Fc and 3 µg/ml anti-human IgG-PE antibody (Jackson Immunoresearch, Cat 109-116-098) was preincubated at 4°C for 1 hr. After pre-binding, the mixture was added to lymphocytes from LNs and staining was done for 1 hr on ice. Cells were then stained as normal for FACS analysis.

## In vivo Scart2 antibody labeling

5 µg Scart2 antibody in 100 µl volume of saline was injected s.c, draining inguinal lymph node was analyzed by flow cytometry 1 hr post antibody injection.

## Surgery and photoconversion

The mouse was anesthetized with ketamine, shaved and antiseptically prepared with 0.02% chlorhexidine gluconate. The mouse was then draped and a ~1.5 cm incision was made in the abdominal skin to expose the left inguinal LN. A silver LED 415 (Prizmatix), set to maximum intensity, with a high numerical aperture polymer optical fiber (core diameter, 1.5 mm) light guide and fiber collimater, was used as a 415 nm violet light source. During the 15 min exposure period, the tissue was kept moist with saline. After photoconversion, the skin was closed with two autoclips (Thermo Fisher Scientific). ~0.1 mg/kg buprenorphine in saline was given i.p immediately before and after surgery, and every 4–12 hr as needed thereafter. The mice were closely monitored for signs of pain. Mice were analyzed immediately before and after photoconversion, and 24 hr and 48 hr post photoconversion. Flow cytometry was used to analyze left and right inguinal LN cells as previously described (*Gray et al., 2013*). The staining was done with antibodies against Vγ4, IL7Rα, Ccr6 and TCRβ.

## Parabiosis

Parabiosis surgery followed previously described procedures (*Smith et al., 2015*). Mirror-image incisions at the left and right flanks were made through the skin and shorter incisions were made through the abdominal wall. The peritoneal openings of the adjacent parabionts were sutured together. Elbow and knee joints from each parabiont were sutured together and the skin of each mouse was stapled (9 mm Autoclip, Clay Adams) to the skin of the adjacent parabiont. Each mouse was injected subcutaneously with Baytril antibiotic and Buprenex as directed for pain and monitored during recovery. For overall health and maintenance behavior, several recovery characteristics were analyzed at various times after surgery, including paired weights and grooming behavior. Mice were sacrificed for flow cytometric analysis two weeks post parabiotic surgery.

## Intravital two-photon laser-scanning microscopy of popliteal LNs

Mice were anaesthetized by intraperitoneal injection of 10 ml/kg saline containing xylazine (1 mg/ml) and ketamine (5 mg/ml). Maintenance doses of intramuscular injections of 4 ml/kg of xylazine (1 mg/ml) and ketamine (5 mg/ml) were given approximately every 30 min. To image the popliteal LN, the mouse's hind leg was immobilized using thermal putty to a Biotherm stage warmer at 37°C (Biogenics) for the duration of the surgery and subsequent imaging. A small incision was made directly behind the knee, and a ~0.5 cm square region of underlying tissue was exposed. The fat pad surrounding the popliteal LN was carefully dissected away without damaging surrounding vasculature and the afferent and efferent lymphatic vessels. After visualization of the popliteal LN, a 3D-printed plastic tissue mount was attached to the surrounding tissue using Vetbond. The tissue mount was immobilized with additional thermal putty, and the area above the LN was submerged in PBS for imaging. Images were acquired with ZEN2009 (Carl Zeiss, Germany) using a 7 MP two-photon

microscope (Carl Zeiss) equipped with a Chameleon laser (Coherent). For video acquisition, a series of planes of 3 µm z-spacing spanning a depth of 90 µm were collected every 30 s. Excitation wavelengths were 905 nm. Emission filters were 500–550 nm for GFP, 570–640 nm for PE, and 450–490 for second harmonic signal. Videos were made and analysed with Imaris 7.4.2 × 64 (Bitplane). Two hours prior to all imaging experiments, 2 µg CD11b-PE or 3 µg MOMA1-TxRed antibody was injected through footpad. Cell tracking was performed using Imaris Bitplane software. Tracks generated using the software were manually confirmed and tracks that were a minimum of 5 min in duration were grouped according to whether they were exclusively in the parenchyma, exclusively in the SCS or crossed between compartments. Each movie contained a total of between 100–300 tracks in the region of interest. Approximately 15% of the Imaris generated tracks could not be confirmed as representing the migration path of a single cell and these were excluded. To quantify the depth of cells from the LN capsule in the imaging data, we used a computational procedure. First, a surface object was created in Imaris over the LN capsule. The positions of both Cxcr6$^+$ cells and CD11b$^+$ subcapsular macrophages were determined using spots objects in Imaris. The minimum distance between each spot and the capsule was determined using a custom MATLAB (MathWorks) script and the ImarisXT interface. These data were exported into the R programming environment for analysis and plotting (ggplot2 package). The depth of the subcapsular sinus in individual LNs varied, and for each experiment was computationally determined as the peak frequency on a plot of CD11b$^+$ subcapsular macrophage's depth below the LN capsule, minus 2 µm, which agreed across experiments with visual estimates of the subcapsular sinus site. As the sinus size varied across experiments from 15 to 40 µm, for the graphs of cell frequency against depth below the capsule the size of the sinus was normalized across experiments to 20 µm. Only the region within the sinus was normalized. Axis ratio was calculated as the ratio of ellipticity (prolate)/ellipticity (oblate). These dimensions were obtained by first creating a surface object for each cell using Imaris Bitplane software. Final videos were annotated and exported in Premiere (Adobe).

## Transswell migration assay

Lymphocytes from LN were allowed to transmigrate for 4 hr across 5 µm transwell filters (Corning Costar, Corning, NY) toward medium or SDF, CCL20, or S1P and enumerated by flow cytometry as described (*Ramirez-Valle et al., 2015*).

## Immunofluorescence microscopy

From paraformaldehyde-fixed tissue, 7 µm sections were prepared, as previously described (*Gray et al., 2012*). In some cases, sections were fixed by acetone (*Gray et al., 2012*). Sections were stained with the flowing antibodies: anti-Lyve1-A647, anti-CD3ε-bio (clone 145-2C11), Goat anti-CCL20 (AF760), Polyclonal Rabbit anti-GFP (Thermo Fisher Scientific), anti-B220-A647 (RA3-6B2), anti-ICAM1-bio (3E2), anti-IL7Rα-A647 (A7R34), rat anti-scart2, Hamster anti-TCRγδ (GL3), Donkey anti-Goat-biotin Jackson immunoresearch), Goat anti-Armenian Hamster-bio (Jackson immunoresearch), Donkey anti-Rat-bio (Jackson immunoresearch), Anti-biotin-A488 (Jackson immunoresearch), Anti-biotin-Cy3 (Jackson immunoresearch). Images were captured with a Zeiss AxioOberver Z1 inverted microscope.

## Cxcr6-GFP cell enrichment

Isolated LN cells from 2–4 Cxcr6$^{GFP/+}$ mice in complete RPMI media (with 2% FBS) were washed twice and resuspended in 200~400 µl PBS (with 2% FBS). Cells were stained with anti CD62L-bio (clone MEL-14), anti CD19-bio (clone MB19-1), anti CD11c-bio (clone N418), anti NK1.1-bio (clone PK136) for 30 min on ice, washed twice, resuspended in 300 µl PBS (2% FBS) with 10~20 µl anti-Biotin MACS beads (5 µl per mouse) and incubated on ice for 30 min. Cells were then washed with and resuspended in 2 ml MACS buffer (PBS, 2% FBS, 2 mM EDTA, pH 7.2), filtered and loaded onto a MACS column, following manufacturer instructions to negatively select Cxcr6$^{GFP/+}$ cells. FACS was used to check enrichment and yield.

## In vitro adhesion assay

ICAM1 adhesion assay: 96-Well Costar Assay Plates (High binding polystyrene, Corning) were coated with ICAM1 at 10 µg/ml concentration in 0.1 M Na$_2$CO$_3$/NaHCO$_3$ pH 9.5 buffer at 4°C O/N.

Plates were then blocked with 1% BSA in RPMI for 30 min at room temperature. Plates were washed twice with RPMI with 0.5% BSA. One million Lymphocytes were added and adhesion assays performed at 37°C for 30 min. A 200-µl pipette was used to wash cells (3x 12 o'clock, 3x 6 o'clock, 1x 12 o'clock, 3 o'clock, 6 o'clock and then 9 o'clock). After washing, cells were eluted with 5 mM EDTA in RPMI with 0.5% BSA and incubation on ice for 15 min. Cells were stained and analyzed by FACS. CD169-Fc/R97A CD169-Fc adhesion assay: Goat Anti-human IgG antibody (Jackson Immunoresearch) was coated on 96-Well Costar Assay Plate (High binding polystyrene, Corning) at 15 µg/ml in 0.1 M $Na_2CO_3/NaHCO_3$ pH9.5 buffer at 4 degree O/N. After coating, plates were washed twice with PBS. After washing, 0.5 µg CD169-Fc or R97A CD169-Fc in 100 µl PBS was added. After 30 min binding, plates were washed twice with PBS, and then blocked with RPMI with 1% BSA for 30 min at room temp. After blocking, plates were washed twice with RPMI with 0.5% BSA. Enriched $Cxcr6^{GFP/+}$ cells or total LN cells (in RPMI with 0.5% BSA) were added to the plates. Adhesion assays were performed at 37°C for 30 min. After binding, cells were washed once (carefully using a 200 µl pipette to remove the medium) with RPMI with 0.5% BSA solution and adherent cells were analyzed by immunofluorescence microscopy or bright field microscopy for quantification.

## Quantitative RT-PCR

Total RNA from sorted LECs or whole LN was isolated and reverse-transcribed, and quantitative PCR was performed for IL1β, IL23a and CCL20 as described. Data were analyzed using the comparative $C_T$ ($2^{-\Delta\Delta Ct}$) method using *Hprt* as the reference.

## Statistical analysis

Prism software (GraphPad) was used for all statistical analysis. Statistical comparisons were performed using a two-tailed Student's t-test. p values were considered significant when less than 0.05.

## Acknowledgements

We thank members of the Cyster laboratory for helpful discussions and comments on the manuscript, Ying Xu for help with QPCR analysis, Gang Wu for help preparing CD169-Fc, K Karjalainen for SCART2 antibody, J Hinnebusch for attenuated Y pestis, and R Proia, S Coughlin, A-K Hadjantonakis and M Tanaka for mice. JGC is an investigator of the Howard Hughes Medical Institute. YZ was supported by the BMS graduate program and HC is a CRI Irvington postdoctoral fellow. This work was supported in part by grant NIH AI074847.

## Additional information

### Funding

| Funder | Grant reference number | Author |
| --- | --- | --- |
| Howard Hughes Medical Institute | | Yang Zhang<br>Theodore L Roth<br>Elizabeth E Gray<br>Hsin Chen<br>Lauren B Rodda<br>Jason G Cyster |
| National Institute of General Medical Sciences | UCSF BMS Graduate Program, T32GM008568 | Yang Zhang<br>Theodore L Roth<br>Elizabeth E Gray<br>Lauren B Rodda |
| National Institutes of Health | NIH AI074847 | Jason G Cyster<br>Yang Zhang<br>Elizabeth E Gray<br>Hsin Chen<br>Lauren B Rodda |
| National Institutes of Health | DP5-OD12178 | Saul Villeda |
| Wellcome Trust | 103744/Z/14/Z | Paul R Crocker |

The funders had no role in study design, data collection and interpretation, or the decision to submit the work for publication.

## Author contributions

YZ, Conception and design, Acquisition of data, Analysis and interpretation of data, Drafting or revising the article, Contributed unpublished essential data or reagents; TLR, EEG, Acquisition of data, Analysis and interpretation of data, Drafting or revising the article, Contributed unpublished essential data or reagents; HC, Acquisition of data, Drafting or revising the article, Contributed unpublished essential data or reagents; LBR, Acquisition of data, Drafting or revising the article; YL, Acquisition of data; PV, SV, Contributed unpublished essential data or reagents; PRC, Drafting or revising the article, Contributed unpublished essential data or reagents; JGC, Conception and design, Analysis and interpretation of data, Drafting or revising the article, Contributed unpublished essential data or reagents

## Author ORCIDs

Yang Zhang, http://orcid.org/0000-0003-4355-9755
Theodore L Roth, http://orcid.org/0000-0002-3970-9573
Jason G Cyster, http://orcid.org/0000-0003-2345-629X

## Ethics

Animal experimentation: Animals were housed in a specific pathogen-free environment in the Laboratory Animal Research Center at the University of California San Francisco (UCSF), and all experiments conformed to the ethical principles and guidelines approved by the UCSF Institutional and Animal Care and Use Committee, protocol approval number: AN107975-02.

# Additional files

## Supplementary files

• Source code 1. Matlab and R source code files used for computational analysis. Three files are enclosed with the Matlab and R code used for the compuational analysis in the present paper: 1) spotsMinDistanceToSurface.m - Matlab File encoding an ImarisXT extension that can be used to find the minimum distance between a spot object and a surface object (used in the present paper to find the minimum distance between cells and the lymph node capsule); 2) SCSAnalyzeR_Core.R - R script containing code to import raw Imaris track lists (including depth statistic added by spotsMinDistanceToSurface.m), preform data processing, and create the plots displayed in the main and supplementary figures; 3) SCSAnalyzeR_Core_Functions.R - R script containing functions called by the main SCSAnalyzeR_Core.R script

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
