## [Decision Letter]

Thank you for submitting your article "Mechanism of innate-like lymphocyte surveillance of the subcapsular sinus to provide lymph node barrier immunity" for consideration by *eLife*. Your article has been reviewed by two peer reviewers, and the evaluation has been overseen by a Reviewing Editor and Michel Nussenzweig as the Senior Editor. The reviewers have opted to remain anonymous.

The reviewers have discussed the reviews with one another and the Reviewing Editor has drafted this decision to help you prepare a revised submission.

Summary:

In the present study Zhang et al. study the mechanisms that allow IL17-producing innate-like lymphocytes (ILL) to enter, stay within, and egress from the LN SCS. The main findings are that CCR6-CCL20 attract ILL towards the SCS area, that the interaction of S1P1R1 with S1P1 mediates their egress into the SCS lumen, that the interaction of unidentified lectin-binding partners on ILL with CD169 on macrophages is important for their retention in the SCS and that the interaction of ILL LFA-1 with ICAM1 guides ILL out of the SCS and back to the LN parenchyma. Some of the major points emphasized by the authors are not novel (localization of ILL at the SCS for rapid pathogen encounter and pre-committed IL17 production by these ILL upon bacterial infection), having been established by the authors in previous studies or by other groups as cited and discussed. The main extension of previous findings involves the demonstration of a critical role for CCR6 in the localization of innate lymphocytes at the SCS and the provision of evidence that in the absence of CCR6 the innate T lymphocytes respond less well to bacterial challenge. These latter data provide some of the first direct evidence for a functional connection between micro-anatomical localization and functionality of lymphocytes regarding pathogen defense. While this relationship has been invoked as an interpretation of prior results, a direct demonstration of this relationship as provided here is important.

This said, the manuscript requires some changes / additions to improve the presentation and to validate some of the key conclusions.

Essential revisions:

1) The critical points regarding the role of CCR6 in ILL positioning and function require further support by plotting the frequency distribution vs. distance to the LN capsule for CCR6 GFP/+ vs CCR6 GFP/GFP mice (as done for CD169, FT720 and αL). As presented now, the impact of CCR6 is not completely clear – Figure 4—figure supplement 3 are not fully convincing – Figure 2—figure supplement 3/B should also show a bi-lobed iLN for optimal comparison as 3/A. The second approach (Thy1 labeling) does not show statistical significance between KO and Het mice (CCR6) regarding all innate-like T cells (Figure 4), leaving doubt regarding the overall strength of the observed effects.

2) The exact roles of S1P as well as ICAM and CD169 are not quite clear and the functional impact of these molecules on the immune response (e.g. IL17 production) has not been tested. The supplementary data analyzing the positioning of innate-like T cells on tissue sections (Figure 5—figure supplement 2, Figure 6—figure supplement 2, Figure 7—figure supplement 4) should be shown in the main figures (plotting of the frequency distribution vs. distance to LN capsule). This will also provide the reader with a better idea of the overall effects of the experimental perturbations.

3) The authors' suggestion that the interaction of LFA1 and ICAM1 allows ILL to egress from the SCS and home back to the LN parenchyma is not convincingly established in this manuscript. The authors observe a relative enrichment of innate lymphocytes between 10-20µm from the SCS, while fewer cells are directly at the SCS when blocking αL (Figure 7—figure supplement 1, Figure 7—figure supplement 2, Figure 7—figure supplement 3, Figure 7—figure supplement 4). ICAM is highly expressed by SCS Mø and parenchymal cells. Therefore the conclusion that LFA1/ICAM1 controls the "access" to the parenchyma is not fully supported – the authors do not discriminate between retention signals at SCS vs. parenchyma. Parenchymal cells could also provide the critical retention signals – in this scenario αL blockade would also lead to accumulation of innate-like T cells closer to the SCS. Further, Sixt and colleagues have shown that translocation of DCs from the SCS into the LN parenchyma occurs independently of integrins. This difference involving established data on integrin-free DC migration vs. presumption of an integrin-counterligand interaction controlling ILL migration across the same endothelial barrier needs further elaboration, or at least comment. Furthermore, if the authors' assumption is right the number of Thy1-PE positive ILL should continuously increase over time in the SCS of aL-treated mice or should increase in absolute cell counts in the blood of these animals. The authors could easily address this issue experimentally.

---

## [Author Response]

*Summary:*

*In the present study Zhang et al. study the mechanisms that allow IL17-producing innate-like lymphocytes (ILL) to enter, stay within, and egress from the LN SCS. […] While this relationship has been invoked as an interpretation of prior results, a direct demonstration of this relationship as provided here is important.*

*This said, the manuscript requires some changes / additions to improve the presentation and to validate some of the key conclusions.*

We thank the reviewers for their careful review and critique of our manuscript. We appreciate their recognizing the novelty of the findings regarding CCR6-CCL20. However, we believe that the reviewers have understated the novelty of several additional findings in the study. To our knowledge, this work is the first to show that this ILL population promptly produces IL17 in response to bacterial and fungal challenge; that ILLs move in and out of the SCS while remaining resident in the LN; and that this shuttling behavior requires S1PR1, CD169 and LFA1. Furthermore, we are not aware of any previous study showing that CD169 can function to support adhesion of cells under conditions of fluid flow and presumed exposure to shear stresses.

*Essential revisions:*

*1) The critical points regarding the role of CCR6 in ILL positioning and function require further support by plotting the frequency distribution vs. distance to the LN capsule for CCR6 GFP/+ vs CCR6 GFP/GFP mice (as done for CD169, FT720 and αL). As presented now, the impact of CCR6 is not completely clear – Figures 4S3 A/B are not fully convincing – S3/B should also show a bi-lobed iLN for optimal comparison as S3/A. The second approach (Thy1 labeling) does not show statistical significance between KO and Het mice (CCR6) regarding all innate-like T cells (Figure 4), leaving doubt regarding the overall strength of the observed effects.*

As requested, we have now quantitated the number of cells proximal to the capsule (within 100µm) versus more distant. This was done by counting cells in tissue sections rather than with intravital 2-photon microscopy because the high CCR6 expression in B cells necessitated costaining with a B cell marker to distinguish CCR6^hi^ ILLs from B cells. We find that there is a ~3-fold reduction in the fraction of KO cells that are proximal to the capsule versus more distant (Figure 4—figure supplement 3). We have replaced Figure 4—figure supplement 3 with a bi-lobed iLN as recommended. We also now include quantitation data for the Scart2 stained sections and this demonstrates a significant reduction in the fraction of these γδ T cells that are adjacent to the sinus (Figure 4—figure supplement 4). These data are in accord with the Thy1-PE labeling data showing that the frequency of IL7Rα^hi^CCR6+ cells and Vγ4+CCR6+ cells in the sinus is significantly reduced compared to WT mice.

*2) The exact roles of S1P as well as ICAM and CD169 are not quite clear and the functional impact of these molecules on the immune response (e.g. IL17 production) has not been tested. The supplementary data analyzing the positioning of innate-like T cells on tissue sections (Figure 5—figure supplement 2, Figure 6—figure supplement 2, Figure 7—figure supplement 4) should be shown in the main figures (plotting of the frequency distribution vs. distance to LN capsule). This will also provide the reader with a better idea of the overall effects of the experimental perturbations.*

We have performed experiments examining the IL17 response of FTY720 treated, and anti-CD169 and anti-LFA1 treated mice challenged with *S. aureus* bioparticles. In contrast to the findings with CCR6 deficiency, we do not detect a significant effect on the IL17 response of these perturbations and this is now stated in the text. We appreciate the suggestion to move Figure 5—figure supplement 2, Figure 6—figure supplement 2 and Figure 7—figure supplement 4 into the main figures but due to the already crowded nature of these figures we felt it made for easier viewing to leave these panels in their (readily accessible) supplementary location. We do now include data for manually confirmed tracks in the main figures and these results are in close accord with the frequency distribution data (new Figure panels Figure 3, Figure 5, Figure 6).

*3) The authors' suggestion that the interaction of LFA1 and ICAM1 allows ILL to egress from the SCS and home back to the LN parenchyma is not convincingly established in this manuscript. The authors observe a relative enrichment of innate lymphocytes between 10-20µm from the SCS, while fewer cells are directly at the SCS when blocking αL (Figure 7—figure supplement 1, Figure 7—figure supplement 2, Figure 7—figure supplement 3, Figure 7—figure supplement 4). ICAM is highly expressed by SCS Mø and parenchymal cells. Therefore the conclusion that LFA1/ICAM1 controls the "access" to the parenchyma is not fully supported – the authors do not discriminate between retention signals at SCS vs. parenchyma. Parenchymal cells could also provide the critical retention signals – in this scenario αL blockade would also lead to accumulation of innate-like T cells closer to the SCS. Further, Sixt and colleagues have shown that translocation of DCs from the SCS into the LN parenchyma occurs independently of integrins. This difference involving established data on integrin-free DC migration vs. presumption of an integrin-counterligand interaction controlling ILL migration across the same endothelial barrier needs further elaboration, or at least comment. Furthermore, if the authors' assumption is right the number of Thy1-PE positive ILL should continuously increase over time in the SCS of aL-treated mice or should increase in absolute cell counts in the blood of these animals. The authors could easily address this issue experimentally.*

We appreciate the reviewers’ critical assessment of these data and we apologize for our lack of clarity about the αL blocking experiments. We have now performed additional analyses that we hope improve clarity. The data in Figure 6 showed that αL blockade for 6 hr caused a marked accumulation of ILL in the Thy1-PE exposed sinus. Video 4 showed that cells could be observed within the sinus and they exhibited a non-migratory, elongated morphology. In Figure 7 we show that there is an increase in the fraction of ILL that are Thy1-PE exposed after 2hr αL treatment and a further increase after 4 hr.

As a further approach to respond to this concern we have performed a manual confirmation of tracks in each of the treatment conditions and assigned tracks based on whether they remained in the parenchyma, remained in the SCS or moved between parenchyma and SCS. This analysis revealed that αL treatment increased the fraction of tracks that were within the SCS. The 3-6 hr αL treatment did not cause a significant change in the fraction of tracks that crossed from the parenchyma into the sinus but it strongly reduced the fraction of tracks that returned from the SCS to the parenchyma (new Figure 6). These data provide support for the conclusion that LFA1 is required for efficient migration of ILL from the sinus into the parenchyma. However, we agree with the reviewers that we cannot exclude LFA1 also having a role on the parenchymal side and have revised the discussion of these data and now comment on how this finding differs from the findings made regarding requirements for BM-derived DCs to move from the sinus into the parenchyma.

We have added data showing that long-term (3 day) aL blockade causes a reduction of ILL in the LNs and an increase in blood, similar to the findings in ICAM1 KO mice (Figure 6—figure supplement 1). These data are consistent with a role for LFA1 in mediating cell return from SCS to parenchyma, with gradual loss of the cells occurring over time due to lymph flow. However, these data are also consistent with a possible role for the integrin in cell retention within the parenchyma, a concept we acknowledge in the revised discussion.

We have added axis ratio measurements for cells in the SCS of control or αL blocked mice (Figure 6, right panel). The elongated morphology of many of the αL blocked cells is consistent with a reduced ability of these cells to take on a flattened, more adhesive state.

We appreciate the reviewers’ point that ICAM1 is expressed on many cell types and we have revised our remarks to indicate that we do not distinguish which cell type(s) are the necessary source of ICAM1. However, our imaging experiments provide evidence that some relevant cell type(s) must provide lumenally exposed ICAM1. In addition to LECs, this might involve CD169+ SCS macrophages.